# Acetylation regulates ribonucleotide reductase activity and cancer cell growth

Guo Chen[1,8], Yin Luo[2], Kurt Warncke[3], Youwei Sun[1], David S. Yu[1], Haian Fu [2], Madhusmita Behera[4], Suresh S. Ramalingam[4], Paul W. Doetsch[5], Duc M. Duong[6], Michael Lammers[7], Walter J. Curran[1] & Xingming Deng[1]

Ribonucleotide reductase (RNR) catalyzes the de novo synthesis of deoxyribonucleoside diphosphates (dNDPs) to provide dNTP precursors for DNA synthesis. Here, we report that acetylation and deacetylation of the RRM2 subunit of RNR acts as a molecular switch that impacts RNR activity, dNTP synthesis, and DNA replication fork progression. Acetylation of RRM2 at K95 abrogates RNR activity by disrupting its homodimer assembly. RRM2 is directly acetylated by KAT7, and deacetylated by Sirt2, respectively. Sirt2, which level peak in S phase, sustains RNR activity at or above a threshold level required for dNTPs synthesis. We also find that radiation or camptothecin-induced DNA damage promotes RRM2 deacetylation by enhancing Sirt2–RRM2 interaction. Acetylation of RRM2 at K95 results in the reduction of the dNTP pool, DNA replication fork stalling, and the suppression of tumor cell growth in vitro and in vivo. This study therefore identifies acetylation as a regulatory mechanism governing RNR activity.

[1] Departments of Radiation Oncology, Emory University School of Medicine and Winship Cancer Institute of Emory University, 1365C Clifton Road NE, Atlanta, GA 30322, USA. [2] Department of Pharmacology, Emory University School of Medicine and Winship Cancer Institute of Emory University, 1510 Clifton Rd. NE, Atlanta, GA 30322, USA. [3] Department of Physics, Emory University, 400 Dowman Drive, Atlanta, GA 30322, USA. [4] Department of Hematology and Medical Oncology, Emory University School of Medicine and Winship Cancer Institute of Emory University, 1365C Clifton Road NE, Atlanta, GA 30322, USA. [5] Laboratory of Genome Integrity and Structural Biology, National Institute of Environmental Health Sciences, National Institutes of Health, Research Triangle Park, NC 27709, USA. [6] Department of Biochemistry, Emory University School of Medicine, 1510 Clifton Rd. NE, Atlanta, GA 30322, USA. [7] Institute of Biochemistry, Synthetic and Structural Biochemistry, University of Greifswald, Felix-Hausdorff-Str. 4, Greifswald 17487, Germany. [8] Present address: Department of Medical Biochemistry and Molecular Biology, School of Medicine, Jinan University, 510632 Guangzhou, Guangdong, China. Correspondence and requests for materials should be addressed to X.D. (email: xdeng4@emory.edu)

Ribonucleotide reductase (RNR), also known as ribonucleotide diphosphate reductase, is an enzyme that catalyzes the formation of deoxyribonucleotides from ribonucleotides[1]. Deoxyribonucleotides, in turn, are used in the synthesis of DNA. The reaction catalyzed by RNR is strictly conserved in all living organisms[2]. RNR plays a critical role in regulating the total rate of DNA synthesis, so that DNA to cell mass is maintained at a constant ratio during cell division and DNA repair[3]. RNR enzymes are divided into three classes termed class I, class II, and class III, based on how radicals are generated during the reaction[4]. Class I is the most extensively studied RNR and is present in all eukaryotes and some prokaryotes[3]. This subclass of RNR is a heterotetramer composed of two large and two small subunits, RRM1 and RRM2, respectively[5,6]. RRM1 is the regulatory subunit harboring two allosteric sites for its regulation[5–7]. RRM2 generates a stable tyrosine radical which is transferred to RRM1 cysteine residues to initiate the reduction reaction upon binding of the substrate[4,5]. In addition, p53R2 is encoded by the RRM2B gene, is induced by p53, and has been identified as a second radical-providing small subunit in mammalian cells[8]. The major role of p53R2-containing RNR complexes is in regulating the synthesis, replication, and repair of mitochondrial DNA (mtDNA) in non-proliferating cells[9,10].

RNR is allosterically regulated at two levels influencing overall activity and substrate specificity[4,7]. The overall activity is regulated by binding of ATP (stimulatory) or dATP (inhibitory) to the activity site (A site) on the RRM1 subunit[7]. The substrate specificity is regulated by the binding of different types of dNTPs to the specificity site (S site), which is also located on the RRM1 subunit. ATP and dATP increase the reduction of CDP and UDP, whereas dTTP increases GDP reduction, and dGTP upregulates ADP reduction[7]. In addition to allosteric regulation, RNR activity is also tightly regulated during cell-cycle progression and in response to extensive DNA damage[8,11]. RNR activity is restricted in resting cells or cells in the G1 phase[12], and significantly increased as cells commit to DNA replication during the late $G_1$/early S phase[8,11]. In accord with this, RRM2 is expressed exclusively during the late $G_1$/early S phase and is degraded in the late S phase[13]. Cdk1/2-mediated phosphorylation of RRM2 at Thr33 promotes its degradation via cyclin F[11].

A hallmark of cancer is uncontrolled proliferation, which requires sufficient levels of dNTPs. RNR is frequently seen to be deregulated in cancer cells[14,15]. Elevated expression of both RRM1 and RRM2 subunits of RNR occurs in various human cancers, making RNR a potential therapeutic target[14]. Several therapeutic RNR inhibitors, including gemcitabine, clofarabine, and hydroxyurea, are employed clinically to treat a number of cancers[16,17]. Such small molecule RNR inhibitors fall into two classes: nucleoside analogs and redox-active metal chelators, which target RRM1 and RRM2, respectively[14].

Reversible lysine acetylation/deacetylation plays a critical role in regulating many essential proteins involved in diverse cellular processes, including DNA repair, chromatin remodeling, transcription, metabolism, cell survival, and proliferation[18,19]. Here, we report that acetylation of RRM2 at K95 inactivates RNR activity via disruption of RRM2 homodimerization, which in turn acts as a molecular switch to dictate RNR function in response to DNA replication and DNA damage. RRM2 acetylation at K95 suppresses tumor cell growth in vitro and in vivo, and is therefore a potentially attractive strategy for cancer therapy.

## Results

**Acetylation of RRM2 at K95 inactivates RNR.** Protein acetylation confers novel functions on modified proteins, including alterations in subcellular localization[20], binding partners[21],

protein stability[22], and enzymatic activity[23]. To examine whether protein acetylation is involved in the regulation of dNTP synthesis, human lung cancer H1299 cells were treated with two different deacetylase inhibitors, trichostatin A (TSA) that inhibits histone deacetylase (HDAC) classes I, II, and IV, and nicotinamide (NAM) that inhibits sirtuin family members[24], alone or in combination, leading to increased global lysine acetylation in cells (Supplementary Fig. 1a. See complete unedited blots in Source Data file). Intriguingly, combined treatment with TSA and NAM resulted in significantly decreased levels of all four dNTPs (Supplementary Fig. 1b, c), indicating that the dNTP synthesis pathway may be disturbed by protein acetylation. Because RNR is the rate-limiting enzyme required for de novo synthesis of dNTPs[25], RNR enzyme activity was measured following treatment of various cell lines with TSA and NAM as we described previously[25]. Treatment with TSA and NAM significantly reduced RNR activity in normal lung epithelial (i.e., BEAS-2B and HBEC3) and human lung cancer cell lines (i.e., H1299 and H460) (Fig. 1a, b). To test whether the reduction in RNR activity and dNTP pool size following TSA/NAM treatment affects DNA replication fork progression, H1299 cells were treated with TSA and NAM, followed by measurement of replication dynamics employing single-molecule DNA fiber analysis. TSA and NAM significantly slowed DNA replication fork progression (Supplementary Fig. 1d), which may occur through inhibition of RNR activity and reduction of dNTPs synthesis. In addition, time course experiments indicate that a reduction in the S-phase population was observed starting from the 18-h time point after NAM + TSA treatment (i.e. 26.3% at 0 h vs. 16.8% at 18 h). The S-phase population was continuously reduced to 4.6% at the 48 -h time point (Supplementary Fig. 1e, f). These results indicate that treatment with NAM and TSA can reduce the S-phase population in a time-dependent manner. Thus, TSA/NAM-induced inhibition of RNR activity may also reduce the cell population in the S phase.

RRM2 is critical for RNR enzymatic activity[5,7]. To assess whether RRM2 is regulated by acetylation, BEAS-2B, HBEC3, H1299, and H460 cells were treated with a combination of two deacetylase inhibitors (TSA and NAM), followed by IP with RRM2 antibody. Acetylation of RRM2 was analyzed by western blot using an acetylated-lysine-specific antibody. Treatment with TSA and NAM significantly enhanced RRM2 acetylation, but did not affect RRM2 protein levels (Fig. 1c; Supplementary Fig. 1g). To further measure the percentage of acetylated RRM2 (Ac-K RRM2) before and after TSA/NAM treatment in cells, we used Ac-K immunoaffinity beads to deplete Ac-K proteins, including Ac-K RRM2, from cell lysates isolated from H460 and BEAS-2B cells before and after TSA/NAM treatment as previously described[26], followed by western blot analysis of unacetylated RRM2 using anti-RRM2 antibody and quantifying the unacetylated RRM2 on western blot bands using ImageJ software. The percentage of Ac-K RRM2 was calculated using the formula: % Ac-K RRM2 = (total RRM2−unacetylated RRM2)/total RRM2 × 100 as indicated in Supplementary Fig. 2a. To test whether the Ac-K RRM2 can be depleted from lysates by Ac-K immunoaffinity beads, we measured Ac-K RRM2 by IP using Ac-K-specific antibody in the lysates before versus after Ac-K depletion, followed by western blot analysis of Ac-K RRM2 using anti-RRM2 antibody. Before Ac-K depletion, certain levels of Ac-K RRM2 were observed in H460 and BEAS-2B cells, and NAM/TSA enhanced Ac-K RRM2 (Supplementary Fig. 2b). However, no detectable levels of Ac-K RRM2 were observed in the lysates after Ac-K depletion (Supplementary Fig. 2b), indicating a highly efficient depletion of Ac-K RRM2 from lysates by Ac-K immunoaffinity beads. To obtain the percentages of Ac-K RRM2, we measured the total RRM2 and unacetylated RRM2

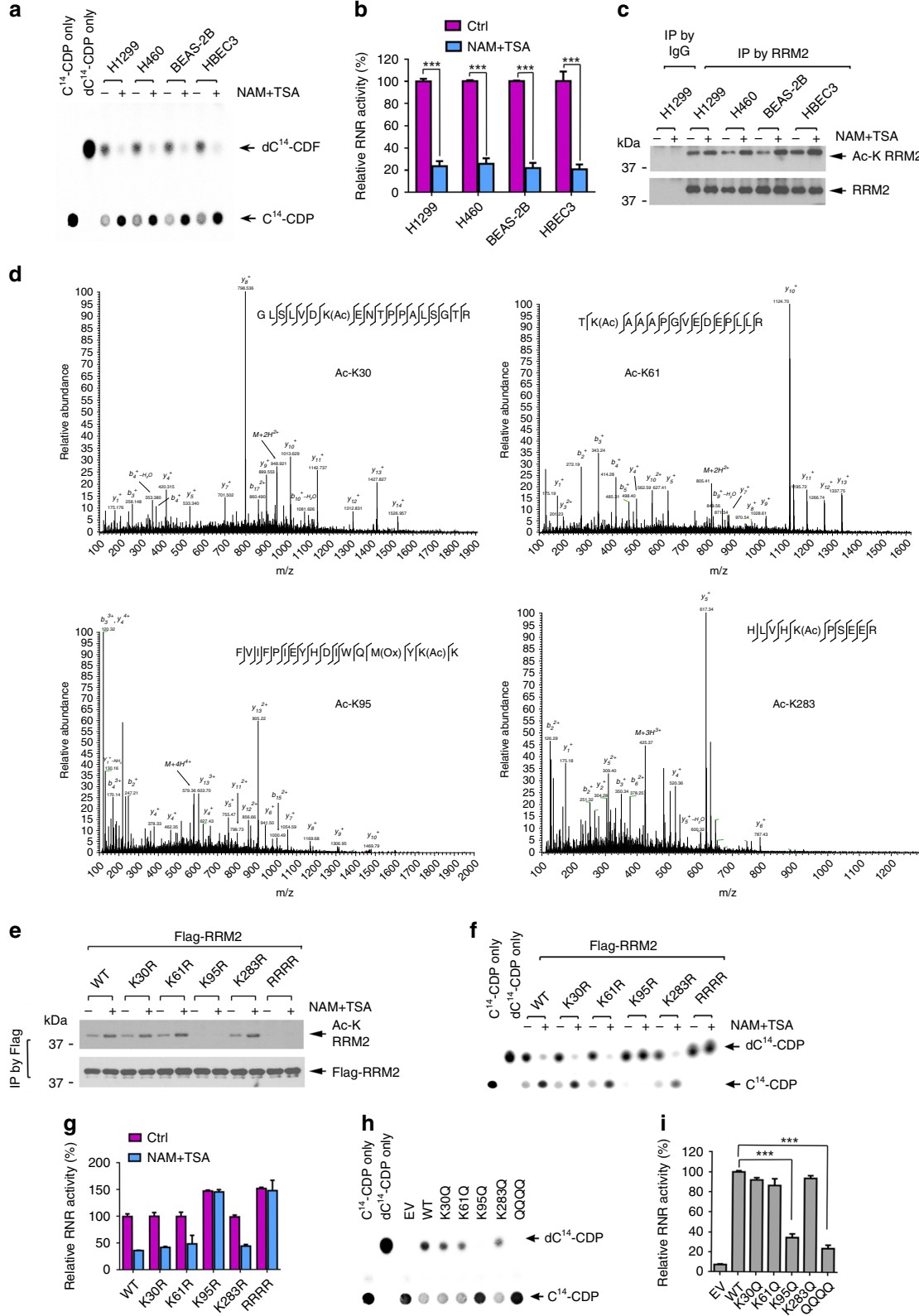

in the lysates before and after Ac-K depletion in H460 and BEAS-2B cells with and without NAM/TSA treatment. We found that 30 and 26% of RRM2 was acetylated in H460 and BEAS-2B cells, respectively, before NAM/TSA treatment (Supplementary Fig. 2c). After NAM/TSA treatment, 76 and 68% of RRM2 was acetylated in H460 and BEAS-2B cells, respectively (Supplementary Fig. 2c).

These results provide more detailed evidence, indicating that NAM and TSA significantly enhance RRM2 acetylation.

To identify the acetylation site(s) of RRM2, liquid chromatography/mass spectrometry (LC/MS) analysis was employed. Four acetylation sites were identified in RRM2, including K30, K61, K95, and K283 (Fig. 1d).

**Fig. 1** RRM2 acetylation at K95 downregulates RNR enzymatic activity. **a**, **b** Extracts from various human cells treated with the combination of NAM (10 mM) and TSA (2 μM) were incubated with $^{14}$C-CDP. The conversion from $^{14}$C-CDP to $^{14}$C-dCDP was analyzed by thin-layer chromatography (TLC). RNR activity was calculated as $^{14}$C-dCDP/($^{14}$C-CDP + $^{14}$C-dCDP). The error bars indicate ± s.d. of three separate experiments. ***$P < 0.001$, by two-tailed $t$ test. **c** Various human cell lines were treated with NAM/TSA for 18 h, followed by IP using an anti-RRM2 antibody. Acetylation of RRM2 (Ac-K RRM2) was analyzed by western blot using acetylated-lysine-specific antibody. **d** H1299 cells were treated with NAM/TSA for 18 h, followed by RRM2 immunoprecipitation and analysis of LC-MS/MS peptide spectra of RRM2 acetylation. **e** H1299 cells expressing Flag-tagged WT or mutant RRM2 were treated with NAM/TSA, followed by Flag IP. RRM2 acetylation was analyzed as above. **f**, **g** Flag-RRM2 variants were immunoprecipitated from H1299 cells treated with NAM/TSA, then mixed with 1 μg of purified GST-RRM1 protein, followed by TLC analysis for RNR activity as above. **h**, **i** The effects of various acetyl-mimetic mutant RRM2 proteins on RNR activity were analyzed as above. The error bars indicate ± s.d. of three separate experiments. ***$P < 0.001$, by two-tailed $t$ test

To determine the role of individual acetylation sites in regulating RRM2 acetylation and RNR activity, we mutated Flag-RRM2 at each of the individual acetylation sites, or at all four sites simultaneously, from lysine (K) to arginine (R) to eliminate acetylation as described[23,27]. This resulted in generation of K30R, K61R, K95R, K283R, and the compound K30R/K61R/K95R/K283R (RRRR) RRM2 mutants. Flag-tagged WT or acetylation-deficient RRM2 mutant(s) were exogenously expressed in H1299 cells. Cells were treated with a combination of NAM and TSA. Flag-RRM2 was immunoprecipitated using a Flag antibody, followed by analysis of acetylation. Substitution of K95 with arginine (K95R) or compound mutations (RRRR) greatly reduced RRM2 acetylation compared to WT (Fig. 1e). Other mutations (K30R, K61R, and K283R) had no significant effect on NAM/TSA-induced RRM2 acetylation compared with WT (Fig. 1e). These findings indicate that K95 is a major acetylation site of RRM2.

To further assess the effect of RRM2 acetylation on RNR activity, a panel of Flag-tagged WT and RRM2 mutant proteins was isolated from NAM/TSA-treated H1299 cells overexpressing exogenous Flag-RRM2, and incubated with recombinant GST-tagged RRM1 protein purified from *E. coli* (Supplementary Fig. 3) and C$^{14}$-CDP, followed by analysis of RNR activity. NAM/TSA reduced RNR activity of WT, K30R, K61R, and K283R, but failed to reduce RNR activity of K95R and RRRR mutant RRM2 proteins (Fig. 1f, g), indicating that K95R and RRRR mutants are deacetylase inhibitor-insensitive phenotypes with sustained enzymatic activity. Replacement of lysine (K) with glutamine (Q) at a protein acetylation site has been reported to mimic lysine acetylation when the function of the acetylation event neutralizes the positive charge of the lysine side chain[27–29]. A series of acetyl-mimetic mutants, including K30Q, K61Q, K95Q, K283Q, and QQQQ, were created by substitution of K with Q. Similar approaches using Flag-tagged acetyl-mimetic RRM2 mutant protein(s) along with recombinant RRM1 and C$^{14}$-CDP were employed to analyze RNR activity. In contrast to the effect of the arginine mutations, the K95Q and QQQQ mutants displayed a marked reduction of RNR activity compared with WT (Fig. 1h, i), indicating that acetylation of RRM2 at K95 led to suppression of RNR activity.

**RRM2 acetylation at K95 disrupts RRM2 homodimerization.** Generation of the stable tyrosyl radical cofactor in RRM2 is essential for RNR catalytic activity[30]. To test whether acetylation of RRM2 at K95 affects tyrosyl radical level, we purified recombinant His$_6$-tagged WT, K95Q, and QQQQ acetylation-mimetic RRM2 proteins from *E. coli* (Fig. 2a). Electron paramagnetic resonance (EPR) spectroscopy reveals that the acetylation-mimetic K95Q and QQQQ proteins have EPR line shapes and peak-to-trough derivative amplitudes for the tyrosyl radical that are comparable with WT (Fig. 2b). Double integration of the derivative-mode EPR spectra shows that the unpaired electron spin concentration is the same in each sample (Supplementary Fig. 4). Therefore, the native tyrosyl radical site structure is not perturbed in the mutant

proteins, and the acetyl-mimetic RRM2 mutant protein is able to generate the same amount of tyrosyl radical cofactor as WT protein. This suggests that RRM2 acetylation-induced suppression of RNR activity may result from mechanism(s) other than its effect on tyrosyl radical cofactor production.

Structural analysis shows that K95 is located at the interface between sister molecules of a RRM2 homodimer, and is predicted to form a salt bridge with Glu174 and Glu105 of the adjacent sister molecule (Fig. 2c). Active RNR is composed of two homodimeric subunits of RRM1 and RRM2[31], and its catalytic center requires both RRM1 and RRM2 homodimers (Fig. 2d)[6]. Thus, RRM2 homodimerization is critical for RNR function. To assess whether K95 acetylation affects RRM2 homodimer formation, HA-tagged WT RRM2 was co-transfected with Flag-tagged acetyl-mimetic RRM2 mutant(s) into H1299 cells. Co-IP experiments showed that Flag-tagged K95Q and QQQQ mutants, but not K30Q, K61Q, or K283Q mutants, were significantly less able to associate with HA-tagged RRM2 compared with Flag-tagged WT (Fig. 2e), indicating that K95 acetylation inhibits dimerization between Flag-RRM2 and HA-RRM2.

To directly evaluate the role of actual acetylation at K95 in regulating RRM2 homodimerization in vitro, in addition to acetyl-mimetic mutant RRM2 K95Q, we employed a genetic-code expansion concept (GCEC) via pRSF-Duet1-MbtRNA$_{CUA}$/AcKRS-3 construct to incorporate acetyl-lysine into RRM2 specifically at the K95 site as recently described[28,29,32]. The pRSF-Duet1-MbtRNA$_{CUA}$/AcKRS-3 construct carries the coding regions for acetyl-lysyl-tRNA-synthetase and the cognate amber suppressor MbtRNA$_{CUA}$ from *Methanosarcina barkeri*[28,29] (Supplementary Fig. 5a). First, human *RRM2* cDNA was cloned into the pRSF-Duet1-MbtRNA$_{CUA}$/AcKRS-3, followed by mutating the K95 gene code from AAG to the amber stop codon TAG that can exclusively be recognized by MbtRNA$_{CUA}$ for acetyl-lysine incorporation. The resulting acetyl K95 *RRM2* (AcK95 *RRM2*) constructs were transformed to *E. coli* BL21(DE3) to produce recombinant AcK95 protein with supplement of N$_\epsilon$-Acetyl-L-lysine (AcK) in the LB medium (Supplementary Fig. 5b). After production of recombinant RRM2 WT, K95Q, AcK95, and K95R proteins, the size-exclusion chromatography (SEC) experiments were performed as described previously[33,34]. Intriguingly, the majority of WT and K95R proteins displayed a homodimeric state while most K95Q and AcK95 RRM2 molecules were detected as monomers (Fig. 2f). Furthermore, purified WT, K95Q, AcK95, and K95R proteins were cross-linked with disuccinimidylsuberate (DSS). Consistent with the SEC data, WT and K95R RRM2 proteins were mainly detected as a dimer, while K95Q and AcK95 RRM2 proteins were mainly detected as a monomer (Fig. 2g). These findings reveal that actual acetylation of RRM2 at K95 (AcK95) or K95 acetyl-mimetic mutation of RRM2 (K95Q) suppresses RRM2 homodimerization.

**KAT7 directly acetylates RRM2 at K95.** To identify the upstream acetyltransferase that acts on RRM2, we used a siRNA library that targets 11 known protein acetyltransferases to screen their effects

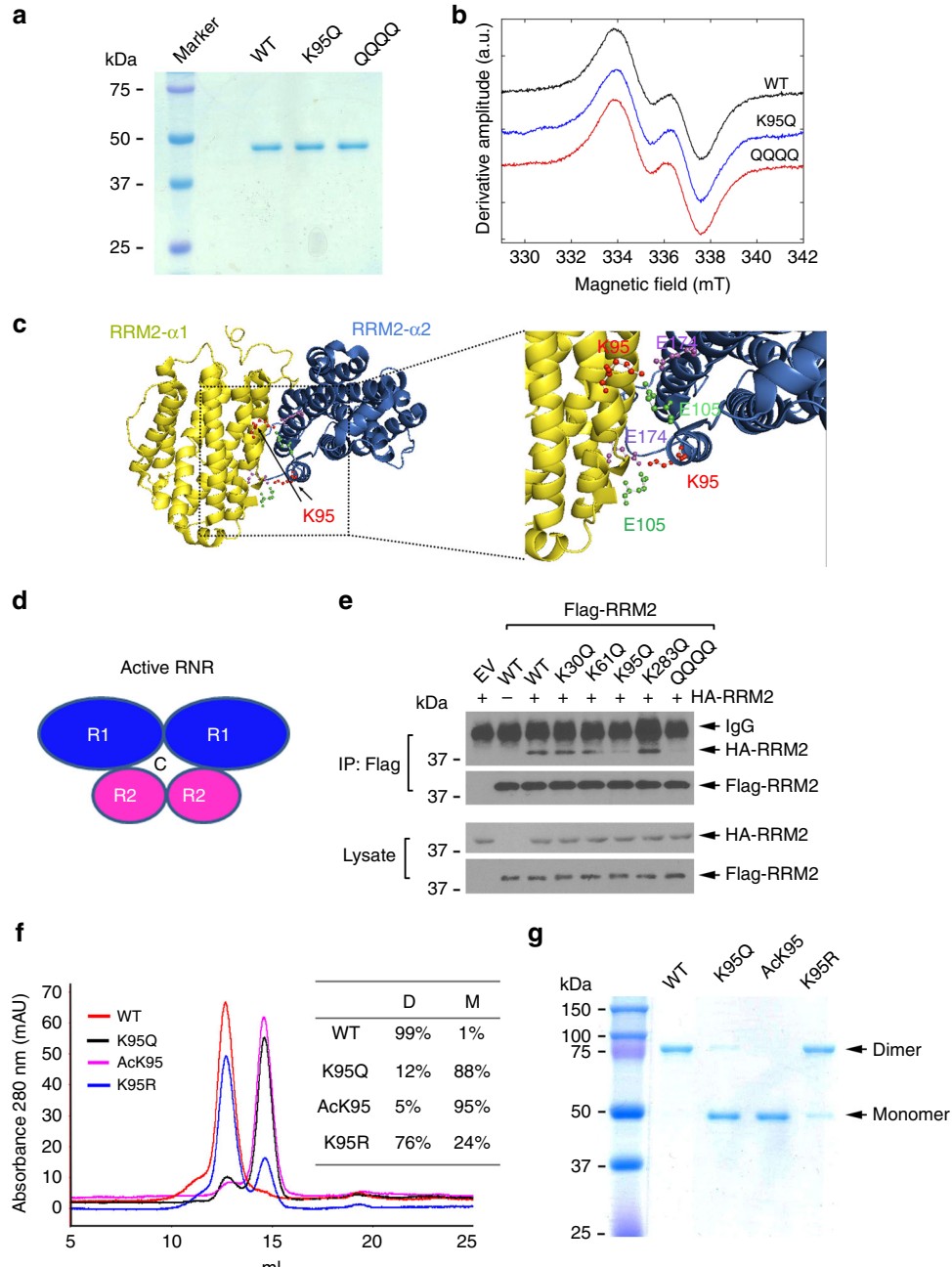

**Fig. 2** RRM2 acetylation at K95 disrupts its homodimerization. **a** Recombinant His-tagged WT, K95Q, and QQQQ RRM2 proteins were purified from *E. Coli*. **b** Tyrosyl radical concentrations were measured by EPR spectroscopy: EPR spectra of WT, K95Q, and QQQQ RRM2 proteins. **c** K95 location within RRM2 structure (left panel). The RRM2 homodimer structure (PDB ID: 3OLJ) is shown, and K95 is predicted to form a salt bridge with E105 and E174 (right panel). **d** Schematic of active RNR complex. **e** H1299 cells were co-transfected with HA-RRM2 and Flag-tagged acetyl-mimetic RRM2 mutants, followed by co-IP using a Flag antibody. HA-RRM2 and Flag-RRM2 were analyzed by western blot using anti-HA or anti-Flag antibody, respectively. **f** Dimerization of recombinant WT, K95Q, AcK95, and K95R RRM2 proteins was analyzed by gel filtration chromatography. **g** SDS-PAGE analysis of recombinant WT, K95Q, AcK95, and K95R RRM2 proteins after cross-linking with 2 mM disuccinimidylsuberate

on RRM2 acetylation (Fig. 3a). Each of these acetyltransferase siRNAs was individually transfected into H1299 cells, followed by analysis of the expression of these acetyltransferases (Supplementary Fig. 6) and RRM2 acetylation (Fig. 3b). Among the 11 siRNAs, only knockdown of KAT7 significantly reduced RRM2 acetylation (Fig. 3b, panel 8), suggesting that KAT7 is an RRM2 acetyltransferase. In further support of this possibility, co-IP experiments showed that KAT7 interacts with endogenous RRM2 (Fig. 3c) or exogenous Flag-RRM2 (Fig. 3d). To assess whether KAT7-induced RRM2 acetylation occurs at K95, *HA-KAT7* was

co-transfected with Flag-tagged WT or K95R mutant *RRM2* cDNA in H1299 cells, followed by analysis of Flag-RRM2 acetylation. HA-KAT7 expression enhanced acetylation of Flag-WT, but not Flag-K95R mutant RRM2 (Fig. 3e). Importantly, purified KAT7 directly acetylated RRM2 WT, but not K95R in vitro (Fig. 3f). Conversely, knockdown of KAT7 using *KAT7* shRNA1 or shRNA2 resulted in reduced RRM2 acetylation (Fig. 3g), enhanced RNR activity (Fig. 3h, i), and elevated levels of dNTPs (Fig. 3j). These findings suggest that KAT7 functions as a physiological RRM2 acetyltransferase.

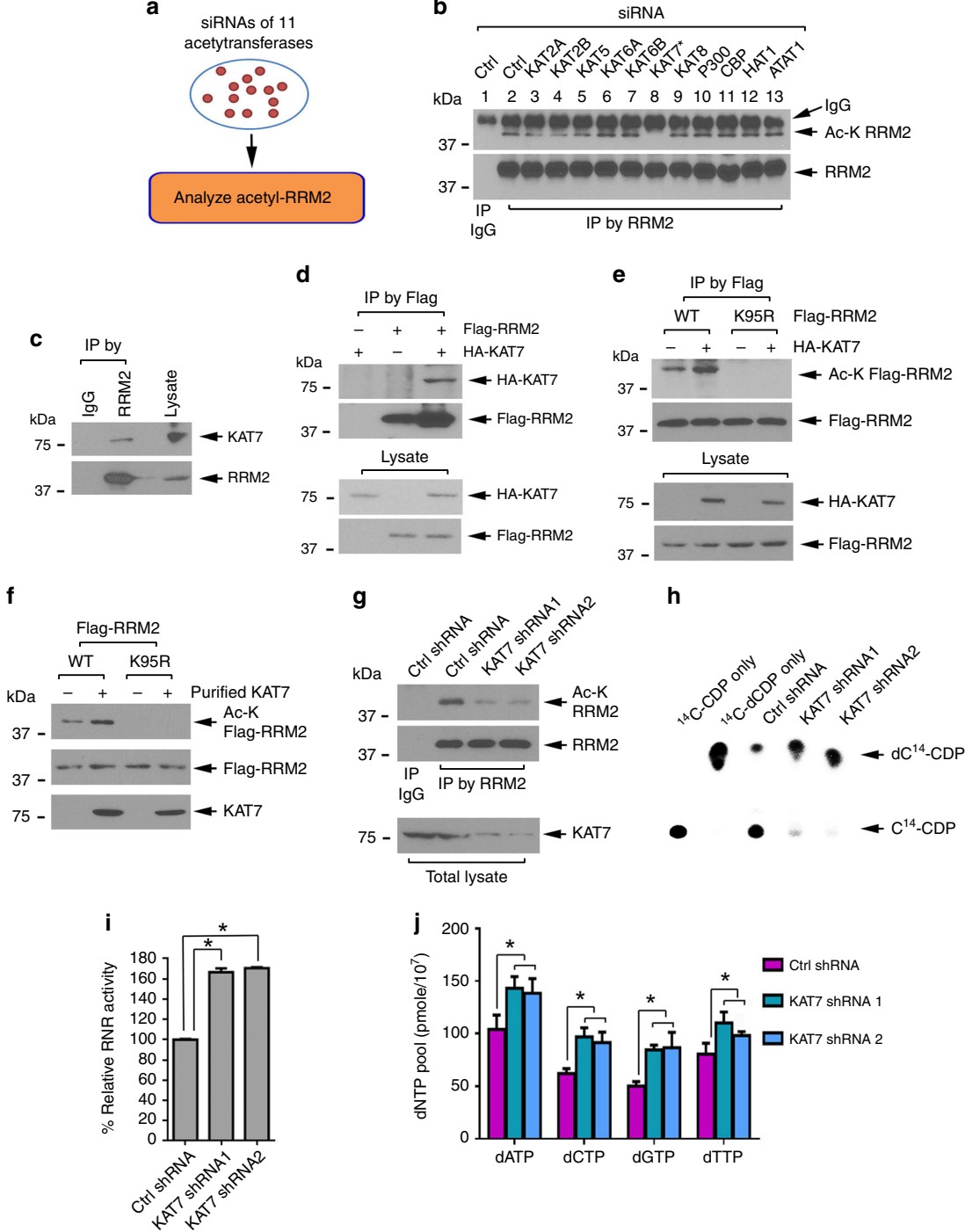

**Fig. 3** KAT7 functions as a physiological RRM2 acetyltransferase. **a** siRNA screening strategy to identify RRM2 acetyltransferases. **b** H1299 cells were transfected with a library of 11 known protein acetyltransferase siRNAs, followed by IP with RRM2 antibody, and western blot using Ac-K antibody. **c** A co-IP was performed in H1299 cells with RRM2 antibody to measure endogenous KAT7/RRM2 interaction. **d** H1299 cells were co-transfected with *Flag-RRM2* and *HA-KAT7* constructs, followed by a co-IP to detect exogenous Flag-RRM2/HA-KAT7 interaction. **e** Co-transfection of Flag-tagged WT or K95R mutant *RRM2* with *HA-KAT7* constructs into H1299 cells, followed by IP using Flag, and western blot using anti-Ac-K antibody. **f** Purified Flag-RRM2 WT or K95R mutant protein were incubated with recombinant KAT7 protein, followed by analysis of RRM2 acetylation as above. **g** Endogenous KAT7 in H1299 cells was depleted by *KAT7* shRNA, followed by analysis of RRM2 acetylation. **h, i** Endogenous KAT7 in H1299 cells was depleted by *KAT7* shRNA, followed by analysis of RNR activity. The error bars indicate ± s.d. of three separate experiments. *$P$ <0.05, by two-tailed $t$ test. **j** Intracellular dNTP levels were measured in H1299 cells expressing *KAT7* shRNA1, *KAT7* shRNA2, and Ctrl shRNA. The error bars indicate ± s.d. of three separate experiments. *$P$ <0.05, by two-tailed $t$ test

**Sirt2 directly deacetylates RRM2 leading to RNR activation.** To identify the physiological deacetylase of RRM2, we used a siRNA library that targets 18 known protein deacetylases (Fig. 4a). Knockdown of these 18 protein deacetylases from H1299 cells was

confirmed by western blot (Supplementary Fig. 7). A increase in RRM2 acetylation was observed only in cells in which Sirt2 was depleted by Sirt2 siRNA (Fig. 4b), suggesting that Sirt2 may be a potential RRM2 deacetylase. To further assess whether Sirt2

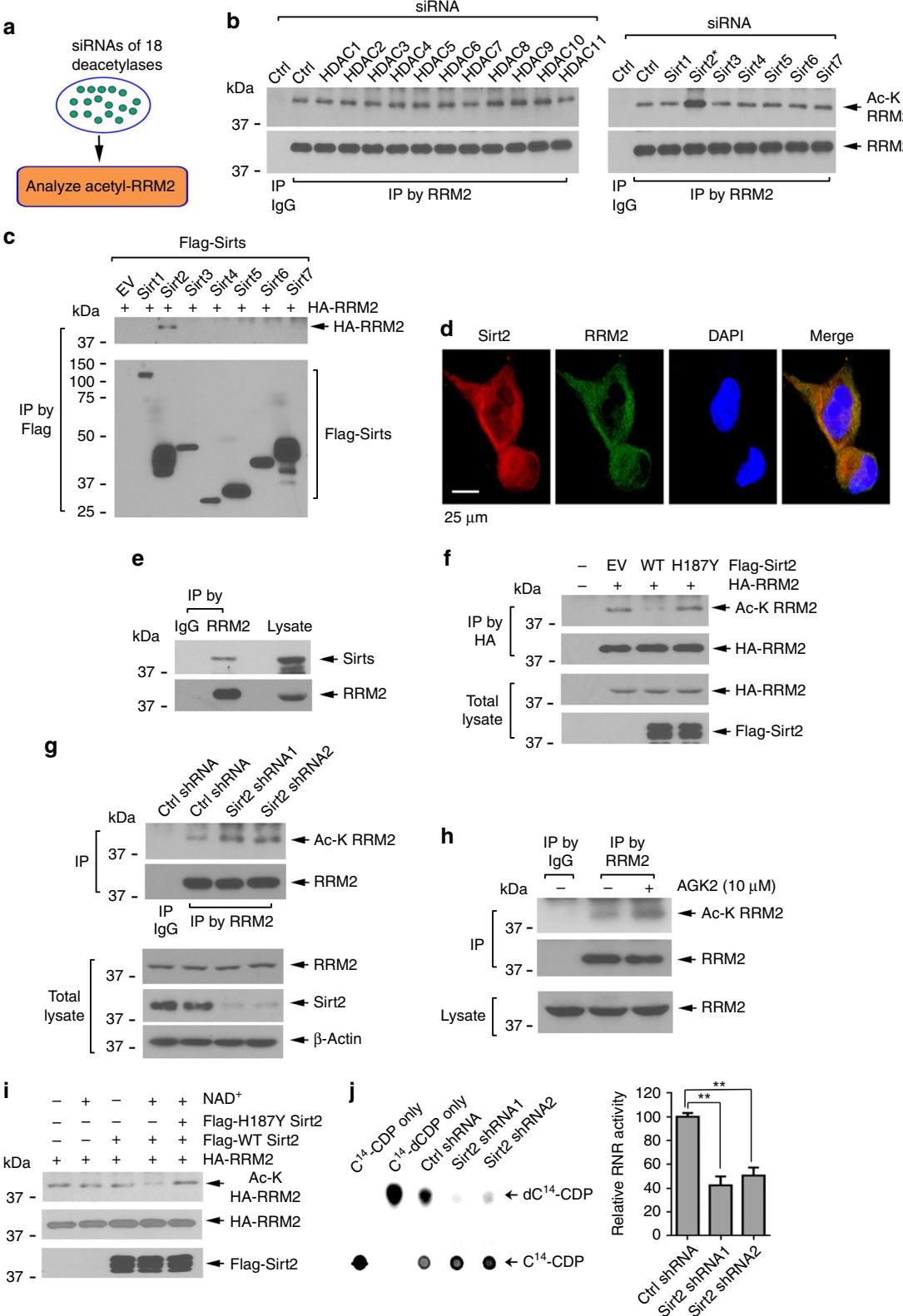

interacts with RRM2, exogenous co-transfection of HA-tagged RRM2 with a series of Flag-tagged sirtuin family members in H1299 cells was performed (Supplementary Fig. 8), followed by co-IP with Flag antibody. Although different sirtuins were overexpressed to different levels, only Sirt2 among seven sirtuin family members was observed to interact with RRM2 (Fig. 4c). Moreover, endogenous

Sirt2 not only co-localized with but also interacted with endogenous RRM2 in H1299 cells (Fig. 4d, e). Intriguingly, significantly increased co-localization of Sirt2 and RRM2 was observed in the S-phase cells compared with the G1-phase cells (Supplementary Fig. 9). The specificity of the Sirt2 antibody used in this experiment was further confirmed as shown in Supplementary Fig. 10.

**Fig. 4** Sirt2 acts as a physiological RRM2 deacetylase. **a** siRNA screening strategy to identify RRM2 deacetylase. **b** H1299 cells were transfected with a library of siRNAs targeting 11 HDAC family members or a library of siRNAs targeting seven sirtuin family members, followed by IP using RRM2 antibody, and western blot with Ac-K antibody. **c** H1299 cells were co-transfected with *HA-RRM2* along with Flag-tagged *Sirt1-7*, followed by co-IP with agarose-conjugated Flag antibody and western blot to detect the Flag-tagged Sirt(s)-associated HA-RRM2. **d** Immunostaining using RRM2 and Sirt2 antibodies in H1299 cells. Scale bar: 25 μm. **e** Co-IP was carried out using RRM2 antibody to detect RRM2/Sirt2 interaction in H1299 cells. **f** H1299 cells were co-transfected with *HA-RRM2* along with Flag-tagged WT or deacetylase-inactive (H187Y) *Sirt2*, followed by analysis of HA-RRM2 acetylation. **g** Sirt2 was depleted from H1299 cells using *Sirt2* shRNA1 or 2, followed by analysis of RRM2 acetylation. **h** RRM2 acetylation was analyzed following treatment of H1299 cells with Sirt2 inhibitor AGK2. **i** HA-RRM2 was isolated from H1299 cells treated with TSA/NAM and then incubated with Flag-WT or H187Y inactive mutant Sirt2 protein purified from 293T cells, followed by analysis of HA-RRM2 acetylation. **j** Sirt2 was depleted from H1299 cells using *Sirt2* shRNA1 or 2, followed by analysis of RNR activity. The error bars indicate ± s.d. of three separate experiments. **P <0.01, by two-tailed *t* test

To test a role of Sirt2 in the deacetylation of RRM2, Flag-tagged *Sirt2* WT or inactive H187Y *Sirt2* mutant[35] was co-transfected with HA-RRM2 into H1299 cells. Intriguingly, overexpression of WT, but not H187Y mutant Sirt2, resulted in a significant reduction in RRM2 acetylation (Fig. 4f). Conversely, depletion of Sirt2 using *Sirt2* shRNA1 or shRNA2 led to a marked increase in RRM2 acetylation (Fig. 4g). Furthermore, treatment of cells with a Sirt2-selective inhibitor AGK2[36] also enhanced RRM2 acetylation (Fig. 4h). To determine if Sirt2 directly deacetylates RRM2 in vitro, acetylated HA-RRM2 was immunoprecipitated from 293T cells overexpressing HA-RRM2 and incubated with purified WT or H187Y mutant Sirt2 in the presence or absence of NAD$^+$ (nicotinamide adenine dinucleotide), a cofactor required for sirtuin deacetylase activity[35]. Purified WT but not H187Y mutant Sirt2 directly deacetylated RRM2 in vitro (Fig. 4i). Functionally, depletion of endogenous Sirt2 from cells led to a significant decrease in RNR activity (Fig. 4j).

The majority of our experiments employed H1299 cells derived from non-small cell lung cancer (NSCLC), therefore, it was of interest to test whether Sirt2 is upregulated in tumor tissues from patients with NSCLC. We analyzed Sirt2 expression in samples from 208 NSCLC patients by IHC staining employing Sirt2 antibody. NSCLC human tissue samples were obtained from the tissue bank at Emory University Winship Cancer Institute. Tissue microarrays (TMA) were generated with replicate cores of tumor and adjacent normal lung. Semiquantitative evaluation of IHC staining of Sirt2 was carried using immunoscores based on both percentage of stained cells and staining intensity, as previously described[37–39]. Sirt2 protein expression was significantly higher in tumor tissues compared with adjacent normal lung tissues (Supplementary Fig. 11a, b). The observed higher levels of Sirt2 could potentially deacetylate RRM2 to increase RNR activity in tumor tissues. Elevated levels of Sirt2 in tumor tissues were correlated with poor outcomes for NSCLC patients (Supplementary Fig. 11c), suggesting that Sirt2, as the RRM2 deacetylase, could be a potential prognostic biomarker for NSCLC. Our findings support and extend those of a previous report that high levels of Sirt2 expression are associated with poor prognosis in NSCLC patients[40].

**Sirt2 regulates a cell-cycle-dependent RRM2 deacetylation.** RNR activity and dNTP pool size are restricted in the G1 phase and expanded in the S phase to ensure sufficient dNTP levels for DNA replication[11,12,41]. To examine the physiological role of RRM2 acetylation/deacetylation in regulating RNR activity during DNA replication, H1299 cells were synchronized at the G1/S boundary using double-thymidine block as previously described[42]. Cells re-entered the cell cycle following release into fresh medium. Most cells entered the S phase at 0–6 h, reached the G2/M phase at 8–10 h, and returned to the G1 phase at 14 h (Supplementary Fig. 12). Notably, RRM2 and its deacetylase Sirt2 but not RRM1 or KAT7 were significantly increased upon entry to the S phase (0–6 h) and

returned to baseline levels in the G1 phase (14 and 16 h), mirroring the expression profiles of the S/G2 marker cyclin A (Fig. 5a). This suggests that elevated Sirt2 in the S phase may deacetylate RRM2 to activate RNR. Correspondingly, RRM2 acetylation was very low in the S-phase cells (0–6 h) and increased in the G1 phase (14 and 16 h) (Fig. 5b, c, left sides), indicating Sirt2-mediated RRM2 deacetylation occurs in the S phase, which would result in increasing RNR activity for the synthesis of dNTPs. In addition, we also used serum starvation as an alternative approach to assess cell-cycle regulation of RRM2 acetylation. H1299 cells were synchronized at the G0/G1 stage by serum starvation, followed by re-addition of serum (10% FBS) to allow cells to re-enter the cell cycle. Cells entered the S phase at 6 h after serum re-addition. Similarly, significant increased levels of Sirt2 in association with decreased levels of RRM2 acetylation were observed in the S/G2 phase (Supplementary Fig. 13). These results demonstrated that Sirt2 regulation of RRM2 deacetylation occurs in a cell-cycle-dependent manner.

To further test the effect of Sirt2 on RRM2 deacetylation during the cell cycle, Sirt2 was inhibited by Sirt2 siRNA or Sirt2-specific inhibitor AGK2. Inhibition of Sirt2 significantly enhanced RRM2 acetylation in the S phase (0–6 h) (Fig. 5b, c, right vs. left sides). These results suggest that Sirt2 is required for the deacetylation of RRM2 in the S phase. To test whether Sirt2 regulates RRM2 deacetylation via K95 during the cell cycle, similar experiments were performed in H1299 cells expressing Flag-tagged WT or the acetyl-deficient K95R RRM2 mutant. The results show that the K95R mutation abrogated cell-cycle-dependent regulation of RRM2 acetylation (Supplementary Fig. 14). Intriguingly, inhibition of Sirt2 using Sirt2 siRNA or Sirt2 inhibitor AGK2 significantly reduced RNR activity (Fig. 5d, e). To assess the effect of Sirt2 on the level of dNTPs, we harvested Sirt2 siRNA- or AGK2-treated H1299 cells and carried out HPLC quantification of dNTPs. Inhibition of Sirt2 by siRNA or AGK2 significantly resulted in a decrease in dNTP pool size (Fig. 5f).

An adequate dNTP pool size is critical to ensure normal cell-cycle progression and cell growth[41]. Employing the BrdU/7-AAD cell-cycle assay, we found that 28.0% of control cells were in the S phase, whereas, silencing of Sirt2 resulted in a reduction of the S-phase cells to 11.3% (Fig. 5g). Similar results were observed following treatment of cells with the Sirt2 inhibitor AGK2 (Fig. 5g). Importantly, disruption of Sirt2 by siRNA or Sirt2 inhibitor significantly suppressed cancer cell growth (Fig. 5h).

To test whether the effects of Sirt2 inhibition can be reversed by the addition of exogenous nucleosides to cells, H1299 cells were transfected with *Sirt2* siRNA, followed by treatment with 20 μM dNTPs for 24 h[43,44]. Cell proliferation was analyzed by BrdU incorporation as previously described[45]. The results show that treatment of cells with dNTPs reverses the inhibitory effect of *Sirt2* siRNA on DNA synthesis and/or cell proliferation (Supplementary Fig. 15).

**RRM2 is deacetylated following DNA damage.** DNA damage is reported to induce RNR activation to ensure adequate dNTP

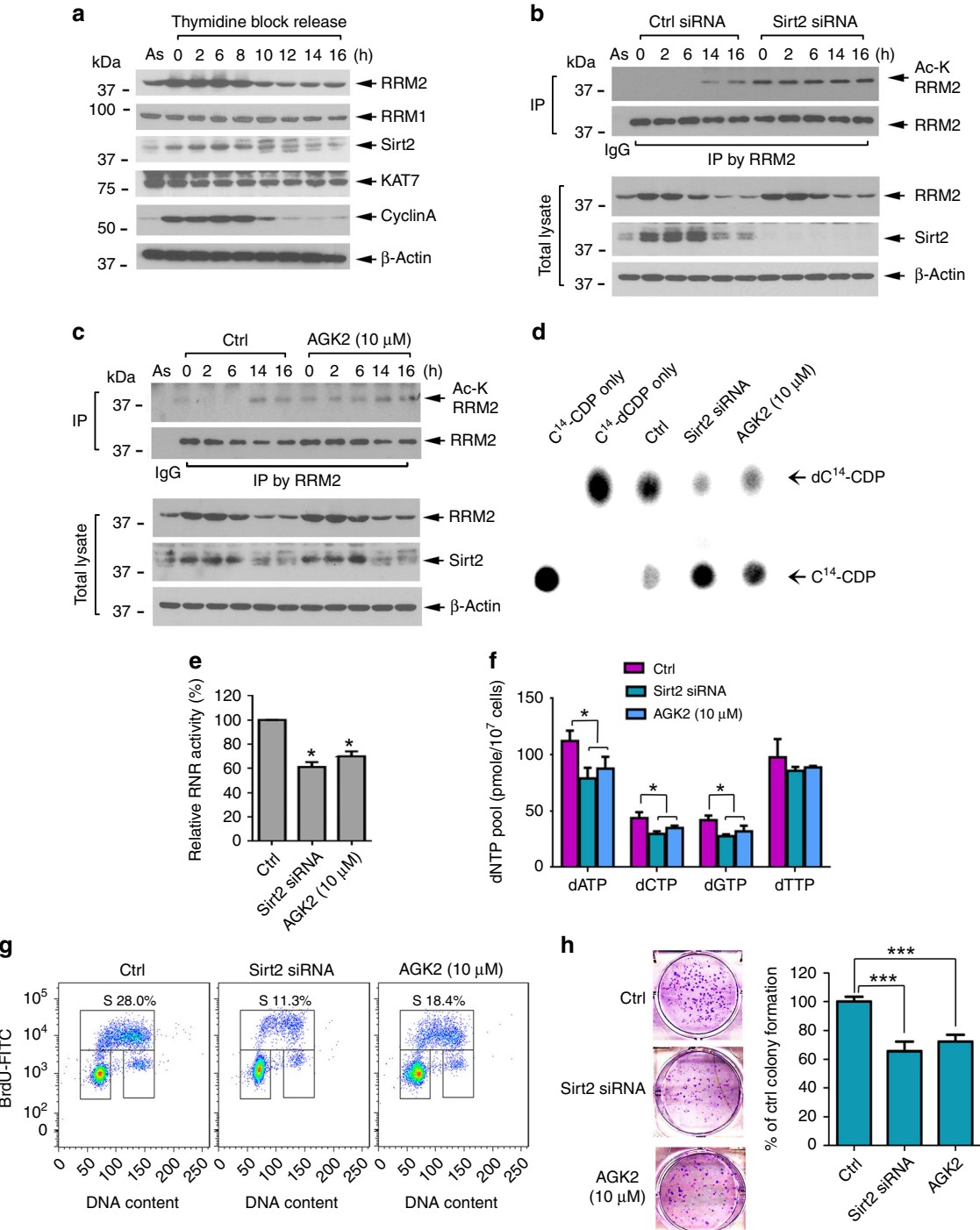

**Fig. 5** Sirt2 deacetylates RRM2 in a cell-cycle-dependent manner. **a** H1299 cells were synchronized at the G1/S boundary by double-thymidine block and then released into normal culture medium. RRM1, RRM2, Sirt2, KAT7, and cyclin A were analyzed by western blot at various time points. As asynchronous. **b**, **c** H1299 cells were transfected with Ctrl or *Sirt2* siRNA or treated with Sirt2 inhibitor AGK2, followed by synchronization, cell cycling, and analysis of RRM2 acetylation as above. **d**, **e** H1299 cells were treated with *Sirt2* siRNA or Sirt2 inhibitor (AGK2), followed by analysis of RNA activity. The error bars indicate ± s.d. of three separate experiments. *P <0.05, by two-tailed *t* test. **f** H1299 cells were transfected with Ctrl or *Sirt2* siRNA or treated with Sirt2 inhibitor AGK2, followed by measurement of dNTP levels. The error bars indicate ± s.d. of three separate experiments. *P<0.05, by two-tailed *t* test. **g**, **h** H1299 cells were transfected with *Sirt2* siRNA or treated with Sirt2 inhibitor AGK2, followed by analysis of cell proliferation and cell-cycle distribution using BrdU/7-AAD staining or colony-formation assay. The error bars indicate ± s.d. of three separate experiments. ***P <0.001, by two-tailed *t* test

levels for DNA repair[8,46,47]. To test whether DNA damage regulates RRM2 acetylation/deacetylation, H1299 cells were treated with ionizing radiation (IR) or camptothecin (CPT), followed by analysis of RRM2 acetylation. A significant decrease in RRM2 acetylation was observed following DNA damage by either agent

(Fig. 6a, upper panel). Intriguingly, IR and CPT exposure also slightly enhanced RRM2 protein expression, but had no effect on KAT7 and Sirt2 protein levels (Fig. 6a, lower panel). Mechanistically, IR and CPT promoted an interaction between Sirt2 and RRM2 (Fig. 6b), but did not influence RRM2/KAT7 association

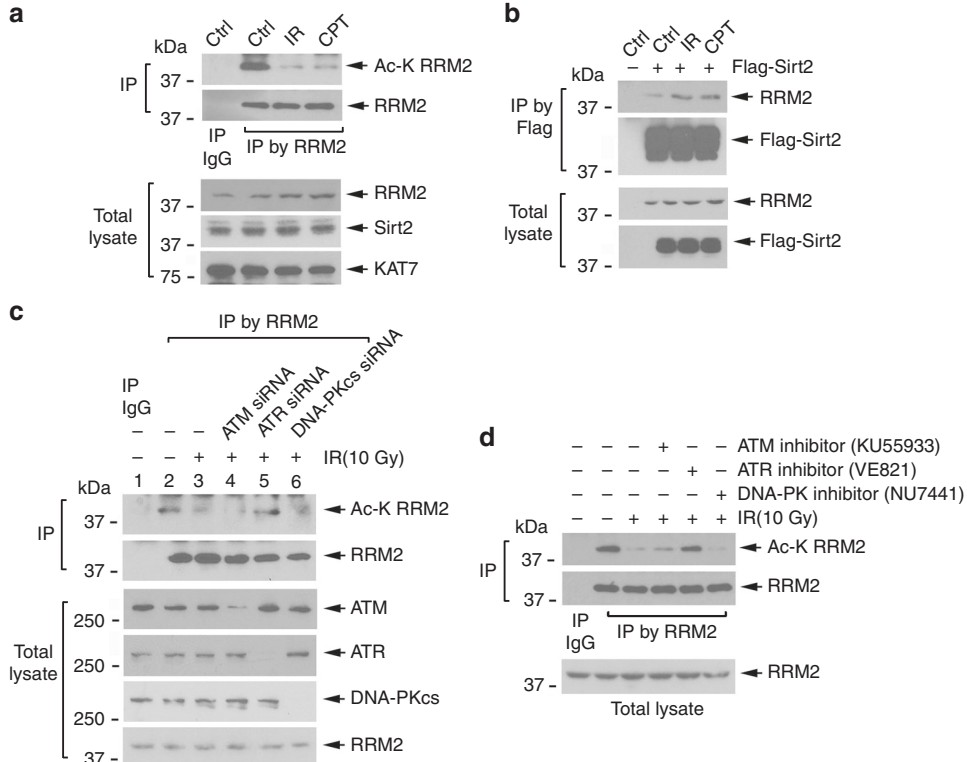

**Fig. 6** RRM2 is deacetylated following DNA damage in an ATR-dependent manner. **a** H1299 cells were exposed to IR (10 Gy) or treated with camptothecin (CPT, 2 μM) for 6 h, followed by IP with RRM2 antibody, and analysis of RRM2 acetylation. **b** H1299 cells expressing exogenous Flag-Sirt2 were treated with IR or CPT, followed by co-IP using Flag antibody. Flag-Sirt2-associated RRM2 was analyzed by western blot using RRM2 antibody. **c**, **d** H1299 cells were transfected with *ATM* siRNA, *ATR* siRNA, or *DNA-PKcs* siRNA or treated with ATM inhibitor (KU55933, 10 μM), ATR inhibitor (VE821, 2 μM), or DNA-PK inhibitor (NU7441, 10 μM), followed by IR, RRM2 IP, and analysis of RRM2 acetylation

(Supplementary Fig. 16), suggesting that DNA damage-reduced RRM2 acetylation mainly results from an increase in Sirt2-mediated deacetylation.

ATM, ATR, and DNA-PKcs play critical roles in DNA damage response signaling pathways[48]. To test whether these kinases affect RRM2 deacetylation in response to DNA damage, siRNA(s) or specific kinase inhibitor(s) for individual kinases were employed. Inhibition of ATR by *ATR* siRNA or the ATR inhibitor VE821 blocked IR-induced deacetylation (Fig. 6c, d). However, suppression of ATM or DNA-PKcs by *ATM* siRNA, ATM inhibitor (KU55933), *DNA-PKcs* siRNA, or DNA-PKcs inhibitor (NU7441) had no significant effect on IR-induced RRM2 deacetylation (Fig. 6c, d). These findings indicate that DNA damage-induced deacetylation of RRM2 is mainly regulated via the ATR signaling pathway.

RRM2 directly interacted with ATR in H1299 cells (Fig. 7a), suggesting that ATR may phosphorylate RRM2 to regulate its deacetylation. It is known that ATR preferentially phosphorylates substrates on SQ/TQ motifs in response to DNA damage[49–51]. To test whether IR or CPT induces RRM2 phosphorylation via ATR, an ATR substrate antibody that specifically recognizes phospho-S/TQ motifs (pS/TQ) was employed. Both IR and CPT stimulated ATR-specific phosphorylation of RRM2 (Fig. 7b). Human RRM2 protein contains only one S/Q motif on serine (S)150 (Fig. 7c). An in vitro ATR kinase assay revealed that purified, active ATR directly phosphorylated WT but not the nonphosphorylatable S150A mutant RRM2 (Fig. 7d), indicating that ATR functions as the physiological kinase that directly phosphorylates RRM2 at the S150 site. Compared with WT or the nonphosphorylatable S150A mutant, the phosphomimetic S150E RRM2 mutant displayed a greater capacity to interact with Sirt2 (Fig. 7e) leading to its

deacetylation (Fig. 7f), indicating that ATR-mediated S150 phosphorylation facilitates Sirt2 deacetylation of RRM2, which results in elevated RNR activity (Fig. 7g, h). IR or CPT stimulated phosphorylation of WT but not S150A or S150E mutant RRM2 (Fig. 7i, j), suggesting that IR- or CPT-induced RRM2 phosphorylation requires the S150 site. Functionally, RRM2-silenced H1299 cells expressing exogenous nonphosphorylatable S150A RRM2 mutant exhibit a higher sensitivity to IR or CPT compared with cells expressing WT or S150E (Fig. 7k, l). These findings suggest that either IR or CPT-induced WT RRM2 phosphorylation or the phosphomimetic S150E RRM2 can increase RRM2 deacetylation leading to RNR activation, which subsequently contributes to decreased sensitivity to IR or CPT.

**Acetylation of RRM2 at K95 stalls DNA replication fork**. Uncontrolled proliferation of cancer cells must be supported by a sufficient dNTP supply, and is reflected by RRM2 overexpression in various types of cancers[14]. Kaplan–Meier survival analysis of 1928 NSCLC patients from an online database (www.kmplot.com) reveals that elevated RRM2 expression is correlated with poor prognosis (Supplementary Fig. 17). To further investigate the functional role of RRM2 acetylation, endogenous RRM2 was depleted from H1299 cells employing *RRM2* shRNA. The effect of shRNAs directed to the 3′-UTR or 5′-UTR of the gene can reportedly be rescued by ectopically expressing the protein using wild-type or mutant cDNA[52,53]. Because the *RRM2* shRNA we employed targets the 3′-UTR of endogenous RRM2, the silencing effect of shRNA on RRM2 expression could be rescued by transfection of exogenous WT or various acetyl-mimetic *RRM2* mutants, as shown in Fig. 8a. Intracellular dNTP levels, DNA replication fork progression, and cell growth were measured in

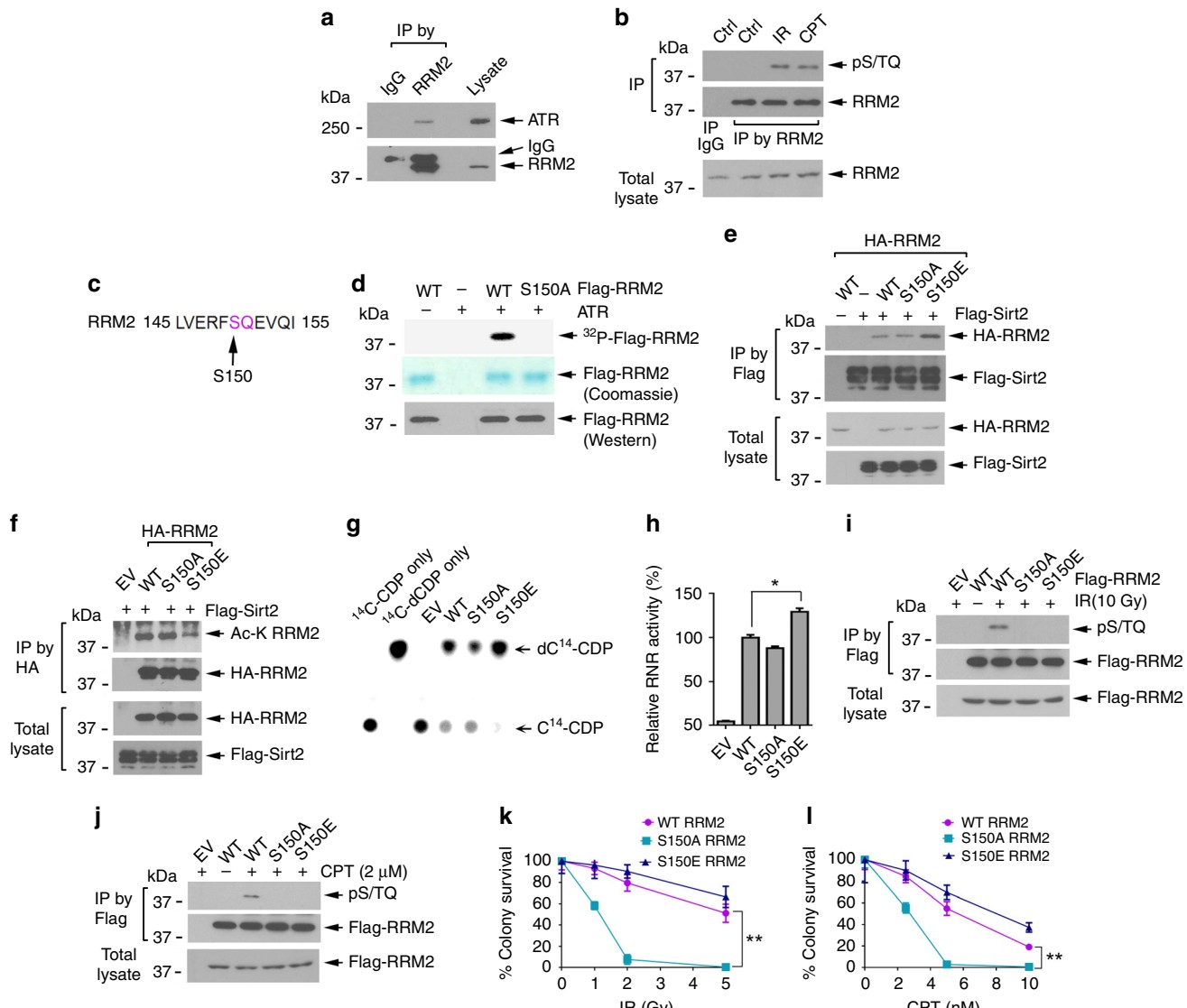

**Fig. 7** ATR phosphorylation of RRM2 facilitates its deacetylation in response to DNA damage. **a** Co-IP in H1299 cells using RRM2 antibody. IgG was used as a IP control. RRM2-associated ATR was analyzed by western blot. **b** H1299 cells were exposed to IR (10 Gy) or treated with CPT (2 μM) for 6 h, followed by IP using RRM2 antibody. Phosphorylation of RRM2 was analyzed by western blot using phospho-S/TQ motif-specific (pS/TQ) antibody. **c** ATR phosphorylation motif "SQ" in human RRM2 protein. **d** Active ATR kinase was immunoprecipitated from H1299 cells treated with IR (10 Gy) and incubated with purified Flag-tagged WT or S150A mutant RRM2 protein in kinase buffer containing [γ-$^{32}$P] ATP. RRM2 phosphorylation was analyzed by autoradiography. **e**, **f** RRM2 phosphorylation at S150 enhances its interaction with Sirt2. Flag-tagged *Sirt2* was co-transfected with HA-tagged WT, S150A, or S150E *RRM2* into H1299 cells, followed by analysis of HA-RRM2/Flag-Sirt2 binding and acetylation of HA-RRM2. **g**, **h** Flag-tagged WT, S150A, or S150E RRM2 mutant protein was immunoprecipitated from H1299 cells overexpressing exogenous Flag-tagged WT, S150A, or S150E, then mixed with 1 μg of purified GST-RRM1 protein, followed by TLC analysis for RNR activity. The error bars indicate ± s.d. of three separate experiments. *$P$<0.05, by two-tailed $t$ test. **i**, **j** H1299 cells overexpressing exogenous Flag-tagged WT, S150A, or S150E mutant RRM2 were treated with IR (10 Gy) or CPT (2 μM), followed by analysis of phosphorylation of Flag-RRM2. **k**, **l** Endogenous RRM2 was depleted from H1299 cells using 3′-UTR targeted *RRM2* shRNA. HA-tagged WT, S150A, or S150E was transfected in H1299 RRM2-deficient cells using pBabe retroviral construct. Cells were treated with IR or CPT, followed by colony-formation analysis. The error bars indicate ± s.d. of three separate experiments. **$P$ < 0.01, by two-tailed $t$ test

RRM2-deficient H1299 cells expressing empty vector, exogenous WT, or individual acetyl-mimetic RRM2 mutants as we described previously[25,54]. Silencing of endogenous RRM2 resulted in a significant reduction in dNTP pool size, stalling of DNA replication fork progression, and suppression of cell proliferation and cell growth. Expression of exogenous WT or K30Q, K61Q, or K283Q but not K95Q or QQQQ RRM2 mutants in RRM2-deficient H1299 cells restored intracellular dNTP pool size, DNA replication fork progression, cell proliferation, and cell growth (Fig. 8a–h).

**Acetylation of RRM2 at K95 suppresses tumor growth**. To further assess whether RRM2 acetylation affects tumor growth, RRM2-deficient H1299 cells expressing exogenous WT or individual acetyl-mimetic RRM2 mutants were employed to establish lung cancer xenografts. Silencing of RRM2 using *RRM2* shRNA significantly inhibited growth of xenografted tumors (Fig. 9a, b, panel 1 vs. panel 2), suggesting that RRM2 is essential for tumor growth. Importantly, expression of exogenous WT, K30Q, K61Q, or K283Q restored tumor growth in the RRM2-deficient xenografts. However, expression of the acetyl-mimetic K95Q or

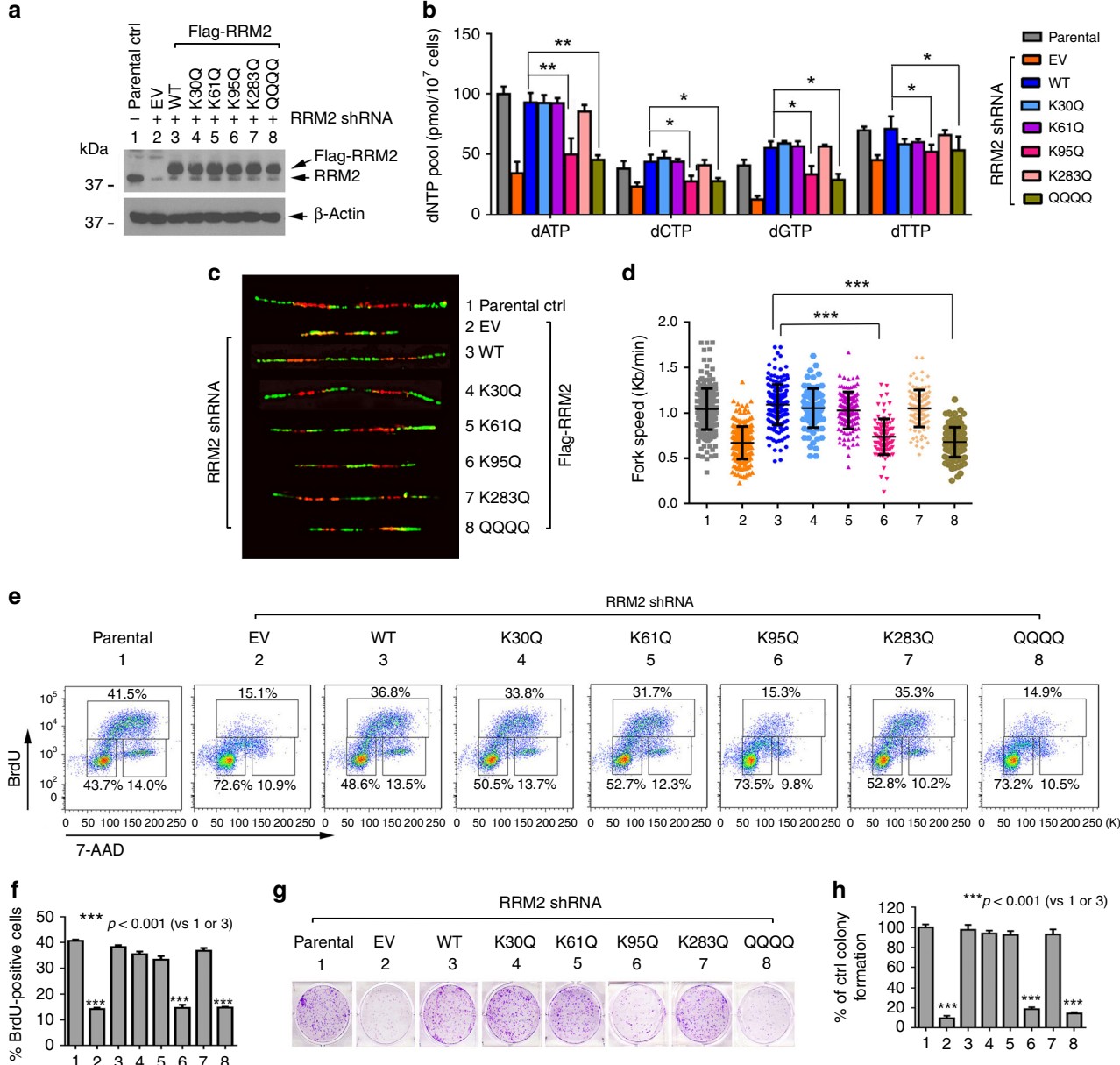

**Fig. 8** RRM2 acetylation at K95 results in reduction of dNTP pool size and growth inhibition. **a** After endogenous RRM2 was depleted from H1299 cells using 3′-UTR-targeted *RRM2* shRNA, Flag-tagged WT or individual acetyl-mimetic RRM2 mutant(s) were exogenously expressed in RRM2-deficient H1299 cells. **b–h** Intracellular dNTPs, DNA replication fork progression, cell proliferation, and cell growth were analyzed by HPLC, DNA fiber assay, BrdU/7-AAD proliferation assay, and colony-formation assay, respectively, in RRM2-deficient H1299 cells expressing empty vector (EV), Flag-tagged WT, or individual acetyl-mimetic RRM2 mutant(s). The error bars indicate ± s.d. of three separate experiments. *$P <0.05$, **$P <0.01$, ***$P <0.001$, by two-tailed *t* test

QQQQ mutant RRM2 failed to rescue tumor growth compared with WT (Fig. 9a, b, panels 6 and 8 vs. panel 3), indicating that RRM2 acetylation at K95 suppresses tumor growth in vivo. In further support of these findings, a significant decrease in Ki67-positive cells was also observed in tumor tissues from K95Q or QQQQ xenografts compared with tumor tissues from WT xenografts (Fig. 9c, d).

## Discussion
Ribonucleotide reductase (RNR) is a unique enzyme that catalyzes the de novo synthesis of dNDPs to provide dNTP precursors required for DNA synthesis[5,6]. Catalysis of NDPs involves a free radical-initiated reduction at the 2′-carbon of ribose 5-phosphate to form 2′-deoxy derivative-reduced 2′-deoxyribonucleoside 5′-diphosphates (dNDPs). An association occurs between the C-terminus of RRM2 and the C-terminus of RRM1, which is required for RNR enzymatic activity[6]. The active site is composed of the active dithiol groups from RRM1, the diferric center and the tyrosyl radical from the RRM2 subunit. Regulation of RNR is designed to maintain balanced levels of dNTPs. In addition to allosteric regulation[7], RNR function is also regulated by SCF$^{cyclin\ F}$ E3 ubiquitin ligase-mediated degradation[11] and small proteins, such as IRBIT[55].

Lysine acetylation is an important post-translational modification known to alter the protein structure and function[56]. The interplay between acetylation and deacetylation is crucial for many important cellular processes[57]. Here, we found that RNR activity is regulated by reversible acetylation/deacetylation of RRM2. Although RRM2 could be acetylated at K30, K61, K95, and K283, K95 was identified as a major acetylation site in the

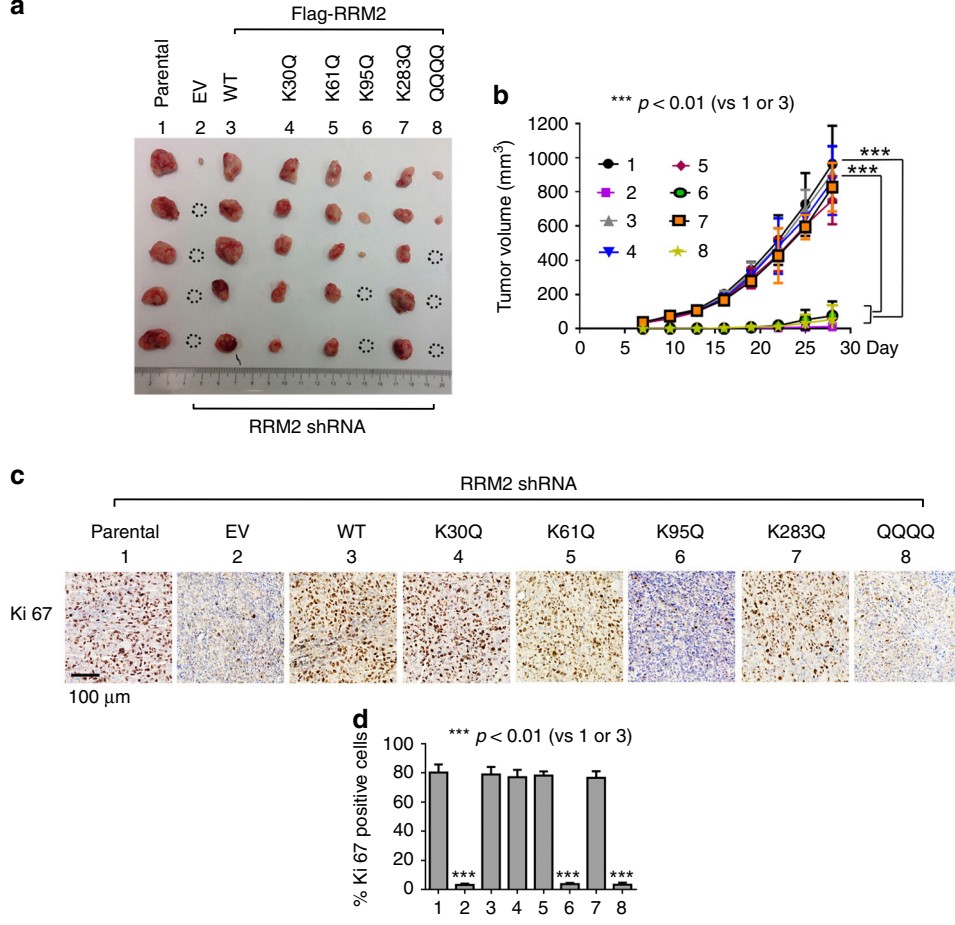

**Fig. 9** RRM2 acetylation at K95 suppresses tumor growth in vivo. **a**, **b** The same number ($3 \times 10^6$) of H1299 parental cells or RRM2-deficient H1299 cells expressing empty vector, Flag-tagged WT, or individual acetyl-mimetic RRM2 mutant(s) were injected into subcutaneous tissue at the flank region of nude mice to generate lung cancer xenografts ($n = 5$ mice each group). Tumor volume was measured once every 4 days. After 28 days, the mice were killed and the tumors were removed, photographed, and analyzed. All dotted circles in tumor images indicate no visible tumors. The error bars indicate ± s.d., $n = 5$ per group. ***$P$<0.001, by two-tailed $t$ test. **c**, **d** IHC staining of Ki67 was performed in tumor tissues at the end of experiments. Scale bar: 100 μm. The error bars indicate ± s.d., $n = 5$ per group. ***$P$ <0.001, by two-tailed $t$ test

RRM2 protein since substitution of K95 with arginine abolished the majority of RRM2 acetylation following NAM/TSA treatment (Fig. 1). Employing acetyl-deficient (K→R) and acetyl-mimetic (K→Q) mutants, we demonstrated that acetylation or deacetylation of RRM2 at K95 inactivated or activated RNR activity, respectively. K95 in RRM2 forms a strong salt bridge with E105 and K95 through its positive charge side chain (Fig. 2c). Thus, the K95Q mutation is able to mimic RRM2 acetylation. It is well known that functionally active RNR enzyme requires formation of an active heterodimeric tetramer which comprises two homodimeric subunits of RRM1 and RRM2[31]. Co-IP and size-exclusion chromatography experiments revealed that K95 acetylation suppressed the ability of RRM2 to form homodimers (Fig. 2), which provides a mechanism by which acetylation inactivates RNR activity.

Reversible acetylation/deacetylation is functionally driven by acetyltransferase and deacetylase enzymes[18,57]. Using siRNA screening strategies, we identified KAT7 as the RRM2 acetyltransferase and Sirt2 as its deacetylase. Depletion of KAT7 by *KAT7* shRNA resulted in a decreased RRM2 acetylation in association with increased RNR activity (Fig. 3). Conversely, knockdown of Sirt2 upregulates RRM2 acetylation leading to decreased RNR activity (Fig. 4). These findings indicate that KAT7/Sirt2-mediated acetylation/deacetylation of RRM2 modulates RNR activity and uncovers a novel regulatory mechanism

of RNR function. Intriguingly, Sirt2 protein levels peak in the S phase, which is associated with RRM2 deacetylation (Fig. 5), indicating that Sirt2-driven RRM2 deacetylation occurs in a cell-cycle-dependent fashion. Since disruption of Sirt2 by siRNA or Sirt2 inhibitor AGK2 suppressed dNTP synthesis leading to cancer cell growth inhibition, pharmacological control of RRM2 acetylation by targeting its upstream deacetylase may represent a novel and effective approach for cancer therapy.

Interestingly, Sirt2 has been reported to have both tumor suppressor[58,59] and oncogenic functions[40,59,60]. On one hand, Sirt2-deficient animals exhibit genomic instability and chromosomal aberrations, and are prone to tumorigenesis[59]. *Sirt2*-knockout mice develop gender-specific tumorigenesis, with females primarily developing mammary tumors, and males more often developing hepatocellular carcinoma (HCC)[58]. Sirt2 functions as a tumor suppressor through its role in regulating mitosis and genome integrity[58]. On the other hand, Sirt2 has also been reported to have tumor-promoting activity[40,59,60]. Sirt2 not only stabilizes Myc oncoproteins to enhance the proliferation of various cancer cell types[61] but also promotes motility and invasiveness of HCC cells by regulating the PKB/GSK-3β/β-catenin signaling pathway[62]. Our and others' findings demonstrated that higher levels of Sirt2 in NSCLC patients are associated with poor prognosis[40] (Supplementary Fig. 11). Moreover, inhibition of Sirt2 by selective Sirt2 inhibitor (thiomyristoyl lysine compound)

exhibits broad anticancer activity[63]. In support of Sirt2's tumor-promoting function, here we discovered that Sirt2 deacetylates RRM2 to enhance RNR activity leading to increased dNTPs pool size and proliferation of cancer cells.

DNA damage is known to stimulate RNR activity to increase levels of dNTPs for DNA repair[8,46,47], but the mechanism(s) involved remain unclear. It has recently been reported that the ATR/Chk1 pathway plays a key role in promoting RRM2 accumulation by stabilizing E2F1 in the S-phase cells, which may be important for countering the replication stress in cancer cells[64]. Here, we discovered that IR- or CPT-induced DNA damage promoted RRM2/Sirt2 interaction leading to RRM2 deacetylation (Fig. 6), suggesting that DNA damage-enhanced RNR activity may also occur through deacetylation of RRM2. ATM, ATR, and DNA-PKcs are major DNA damage response kinases[48]. However, specific inhibition of ATR, but not ATM or DNA-PKcs, blocked IR or CPT-induced RRM2 deacetylation. ATR is mainly located in the nucleus where it executes its DNA damage response function[65]. However, ATR has also been reported to be located in the cytoplasm[66,67]. In contrast, RRM2 is largely cytoplasmic[5], but has also been reported to enter the nucleus during cell-cycle progression or following CPT treatment[11,68]. Based on these reports, ATR and RRM2 could co-localize in the cytoplasm and nucleus. Intriguingly, ATR not only interacts with but also directly phosphorylates RRM2 at S150 (Fig. 7). Thus, ATR-induced RRM2 phosphorylation may occur in both the cytoplasm and nucleus. Functionally, ATR-mediated RRM2 phosphorylation at S150 promotes RRM2/Sirt2 interaction leading to deacetylation of RRM2 and subsequent RNR activation (Fig. 7). These findings reveal a novel mechanism by which IR or CPT-induced DNA damage activates RNR to promote the synthesis of dNTPs for DNA repair. Expression of nonphosphorylatable S150A RRM2 conferred greater cellular sensitivity to IR or CPT, while the phosphomimetic S150E led to resistance compared with cells expressing WT RRM2. This suggests that targeting ATR-mediated RRM2 phosphorylation may represent a potential new therapeutic strategy for cancer therapy.

Cancer cells rely on a sufficient dNTP supply more heavily than normal cells, for their malignant cell growth[15]. Thus, cancer cells have a larger dNTP pool than normal cells[14]. RRM2 is highly overexpressed in various cancers, and this has led to recognition of RRM2 as an effective cancer therapeutic target[14]. Our studies using acetyl-mimetic RRM2 mutants reveal that acetylation of RRM2 at K95 reduces the intracellular dNTPs pool and decreases DNA replication fork progression, leading to suppression of cancer cell growth in vitro and in vivo (Figs 8, 9). These results suggest that K95 acetylation-mediated RNR inactivation can cause DNA replication stress, leading to repression of tumor growth.

In conclusion, our studies demonstrate that acetylation and deacetylation of RRM2 at K95 negatively and positively regulates RNR activity, respectively. RRM2 is acetylated by KAT7 and deacetylated by Sirt2. ATR-mediated RRM2 phosphorylation at S150 enhances its association with Sirt2 and facilitates RRM2 deacetylation. Pharmacological manipulation of K95 acetylation may represent a viable strategy for cancer therapy.

## Methods

**Cell lines and plasmid transfection**. H460, H1299, BEAS-2B, and HBEC3 cell lines were obtained from the American Type Culture Collection. H460 and H1299 cells were grown in the RPMI 1640 medium supplemented with 10% FBS. BEAS-2B cells were cultured in the DMEM/F-12 medium with 10% FBS. HBEC3 cells were cultured in keratinocyte serum-free media from Invitrogen™ (Carlsbad, CA) supplemented with pituitary extract and EGF. These cell lines were tested for mycoplasma and had no mycoplasma contamination. No further authentication for these cell lines was carried out by authors. H1299 cells Flag-tagged WT or

mutant RRM2 mutant(s) in pcDNA3.1 were transfected into H1299 cells using NanoJuice (EMD Millipore) according to the manufacturer's instructions.

**Liquid chromatography coupled to tandem mass spectrometry**. To identify acetylation site(s) on RRM2, Flag-tagged human RRM2 was overexpressed in HeLa cells. Cells were then treated with TSA and NAM for 16 h. Overnight in-gel trypsin digestion was performed on the immunoprecipated Flag-RRM2 resolved on SDS-PAGE gel, and peptides were extracted with a solution of 5% formic acid and 50% acetonitrile and speed vacuumed to dryness. Each peptide sample was resuspended in loading buffer (0.1% formic acid, 0.03% trifluoroacetic acid, 1% acetonitrile), and peptide eluents were separated on a 15 cm 1.9 µm C18 (Dr. Maisch, Germany) self-packed column (New Objective, Woburn, MA) by a Dionex Ultimate 3000 RSLCnano UPLC and monitored on an Orbitrap Fusion Mass Spectrometer (Thermo Fisher Scientific, San Jose, CA). Elution was performed over 80 min gradient at a rate of 300 nl/min with buffer B ranging from 1 to 35% (buffer A: 0.1% formic acid in water, buffer B: 0.08% formic acid in acetonitrile). The mass spectrometer cycle was programmed to collect at the "top speed" mode with a cycle time of 3 s. The MS scans were collected at a resolution of 120,000 (100–1000 m/z range, 400,000 AGC, 50 ms maximum ion time), and the MS/MS spectra were acquired by the ion trap after HCD (high-energy collision dissociation) fragmentation with an isolation width of 2 m/z, 30% collision energy, 10,000 AGC target, and 50 ms maximum ion time. Dynamic exclusion was set to exclude previous sequenced peaks for 20 s within a 10 ppm window. The Proteome Discoverer 2.0 (Thermo Fisher Scientific, San Jose, CA) with the Sequest HT algorithm was used to search and match MS/MS spectra to a complete human uniprotdatabase (downloaded April 2015 with 90270 entries). Search parameters included mass tolerance of precursor ions (± 10 ppm), 0.6 Da for the product ions, fully tryptic restriction, dynamic modifications for deamidated Gln and Asn (+ 0.9840 Da), oxidized Met (+ 15.9949 Da), acetylated lysine (+ 42.0367 Da), static modifications for five maximal modification sites and a maximum of four missed cleavages. The embedded Percolator program was used to filter peptide spectral match (PSM) false discovery rate (FDR) to 1%.

**Site-specific incorporation of Nε-acetyl-L-lysine into RRM2**. We employed a genetic-code expansion concept (GCEC) via pRSF-Duet1-MbtRNA$_{CUA}$/AcKRS-3 construct to incorporate acetyl-lysine into RRM2 specifically at the K95 site as described[28,29,32]. The construct contains the coding regions for acetyl-lysyl-tRNA-synthetase (AcKRS-3) and the cognate amber suppressor MbtRNA$_{CUA}$ from *Methanosarcina barkeri*. First, human RRM2 cDNA was cloned into the pRSF-Duet1-MbtRNA$_{CUA}$/AcKRS-3 between the BamH I and Not I sites. Then, the QuikChange site-directed mutagenesis kit (Agilent Technologies, Santa Clara, CA) was used to mutate RRM2 K95 from AAG to TAG, which is exclusively recognized by MbtRNA$_{CUA}$ for acetyl-lysine incorporation. Primers for mutagenesis: forward, 5′-CAT GAT ATC TGG CAG ATG TAT TAG AAG GCA GAG GCT TCC TTT TGG-3′; reverse, 5′-CCA AAA GGA AGC CTC TGC CTT CTA ATA CAT CTG CCA GAT ATC ATG-3′. After 26 PCR cycles, the amplification product was digested with Dpn I to remove the non-mutated template and transformed into DH5α for amplification. The correct mutation was confirmed by sequencing. The resulting acetyl K95-RRM2 (AcK95-RRM2) construct was transformed into *E. coli* BL21 (DE3) to produce recombinant AcK95 protein. *E. coli* BL21 (DE3) were cultured in the terrific broth medium at 37 °C with shaking at 220 rpm. 10 mM Nε-acetyl-L-lysine and 20 mM nicotinamide were added when OD$_{600}$ reached 0.6. The *E. coli* were grown for another 0.5 h. After addition of 200 µM IPTG, *E. coli* BL21 cells were grown for 16 h at 20 °C. After expression, *E. coli* were then harvested and resuspended in buffer A (20 mM Tris, pH 8.0, 120 mM NaCl, 10% glycerol and 0.5 mM PMSF), followed by sonication. The lysate was centrifuged at 14,000 rpm for 30 min. The resulting supernatant was applied to a Ni-affinity column. After extensively washing with buffer A, bound proteins were eluted with buffer B (20 mM Tris, pH 8.0, 120 mM NaCl, 300 mM imidazole, and 10% glycerol). The eluted proteins were dialyzed with buffer A and stored at −80 °C.

**In vitro deacetylation assay**. 293T cells were transiently transfected with HA-tagged RRM2 and treated with 0.5 µM TSA and 25 mM nicotinamide for 8 h. The acetylated HA-RRM2 protein was immunoprecipitated using agarose-conjugated anti-HA antibody as a substrate. To purify WT or mutant Sirt2 protein, Flag-tagged WT or H187Y mutant Sirt2 was transfected into 293T cells, followed by IP using anti-Flag M2 affinity beads. Flag-Sirt2 protein was eluted with TBS (50 mM Tris-HCL pH 7.6, 150 mM NaCl) supplemented with 2 µg/µl 3 × Flag peptide for deacetylation assay. The acetylated HA-RRM2 protein (300 ng) on agarose beads was incubated with purified Sirt2 WT or H187Y mutant protein (2 µg) in deacetylation buffer (50 mM Tris-HCL pH 8.0, 150 mM NaCl, 1 mM MgCl$_2$, 5 mM NAD$^+$) at 30 °C for 3 h. The reaction was stopped with SDS sample buffer, subjected to SDS-PAGE, and analyzed by western blot with acetylated-lysine antibody.

**Measurement of intracellular dNTPs**. Cellular dNTP levels were analyzed as we described previously[25,69]. Briefly, cells were harvested, and cellular nucleotides were extracted with 6% trichloroacetic acid followed by neutralization with the addition of 5 M K$_2$CO$_3$ just prior to HPLC analysis. The dNTPs were separated from NTPs using a boronic acid resin column (Thermo Fisher Scientific, Waltham, MA). Then,

chromatographic separations of dNTPs were performed using a Symmetry C (18) 3.5 μM (150 × 4.6 mm) column equipped with fluorescence detector (Waters Corporation, Milford, MA).

**Patient samples and immunohistochemical (IHC) staining**. Informed consent was obtained from all human subjects, and use of human samples for IHC was approved by the Institutional Review Board of Emory University. Paraffin-embedded human lung tissue samples from 208 NSCLC patients were obtained from the tissue bank at Emory University Winship Cancer Institute. Tissue microarray (TMA) was constructed with replicate cores of tumor and adjacent normal lung. After deparaffinization, rehydration, inactivation of endogenous peroxidase, and antigen retrieval, IHC staining was performed using R.T.U. Vectastain Kit (Vector Laboratories) according to the manufacturer's instructions. A 1:100 dilution of Sirt2 primary antibody was employed. The semiquantitative evaluation of IHC staining was carried out using immunoscore based on both percentage of stained cells and staining intensity as described[37]. The intensity was defined as follows: 0, no appreciable staining; 1, weak intensity; 2, moderate intensity; 3, strong intensity; 4, very strong intensity. The immunoscore was calculated by multiplying the intensity by percentage of positive staining, producing a total range of 0 to 400. For IHC analysis of Ki67 in tumor tissues from xenografts, Ki67-positive cells in tumor tissues were scored at ×400 magnification. The average number of positive cells per 0.0625 mm$^2$ area was determined from three separate fields in each of three independent tumor samples.

**Size-exclusion chromatography**. Size-exclusion chromatography (SEC) is a gel filtration chromatography, in which proteins in solution are separated by molecule size or weight[33,34]. Gel filtration chromatography was performed using AKTA protein purification system equipped with superdex 75 10/300 column (GE Healthcare, UK). Recombinant RRM2 WT or K95Q protein (1 μg/μl) was loaded on a superdex 75 10/300 column. WT or K95Q protein was eluted from the column with Tris buffer (50 mM Tris-HCl, pH = 8.0, 150 mM NaCl) at 4 ℃ with a flow rate of 1 ml/min. The presence of RRM2 protein in the eluate was detected by monitoring its UV absorbance at 280 nm. The molecular weight and dimer state of RRM2 were determined based on migration of molecular weight standards (ribonuclease A→12 kDa, ovalbumin →44 kDa, conalbumin →75 kDa, r-globulin→158 kDa, thyroglobulin →670 kDa) using a gel filtration marker kit (MWGF200, Sigma, MO).

**Statistical analysis**. All data are presented as mean ± standard deviation (s.d.) from at least three independent experiments. The statistical significance of differences between groups was analyzed with two-tailed $t$ test. We chose the sample size to detect a minimum effect size of 1.5 with at least 80% power and a type I error of 0.05 for each comparison. The log rank test was used to test differences in the Kaplan–Meier survival assay. A value of $P < 0.05$ was considered to be statistically significant. The investigator was not blinded during experiment or assessing the outcome.

**Additional methods**. A list of antibody used and additional methods can be found in the Supplementary Methods section.

**Reporting summary**. Further information on research design is available in the Nature Research Reporting Summary linked to this article.

## Data availability
The data that support the findings of this study, its supplementary information, and source data file are included in the paper, and are available from the corresponding author (X. Deng) upon reasonable request. The data underlying each figure are provided as a source data file. The mass spectrometry proteomics source data are available via ProteomeXchange with identifier PXD014105.

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

## Acknowledgements

This work was supported by NIH/NCI grants R01CA193828, R01CA136534 and R01CA200905–01A1 (to X. Deng), by U01CA217875 (to H. Fu), by 5R01DK054514 (to K. Warncke), by R01CA178999 (to D.S.Y.), by NIH Intramural Research Program Projects Z1AES103328 (to P.W.D.) and Z1AES103266 (to P.W.D.), by the Winship Research Informatics, Pathology and Integrated Cellular Imaging shared resource, the cores supported by the Winship Cancer Institute of Emory University (P30CAJ 38292), by the Winship Fashion a Cure Research Scholar Award (to X. Deng), a philanthropic award provided by the Winship Cancer Institute of Emory University, and by Winship Endowment Fund (to X. Deng). We are grateful to Dr. Michele Pagano (New York University School of Medicine) for kindly providing pcDNA3.1-Flag-RRM2. We thank Anthea Hammond for editing of the paper.

## Author contributions

X.D. conceived the project. G.C., Y.L., K.W., and Y.S. performed experiments. D.M.D. performed LC-MS/MS analysis, D.S.Y. provided technical support for Sirt2 study. S.S.R., M.B., and W.J.C. provided patient samples. M.L. provided pRSF-Duet-1/MbtRNA$_{CUA}$/AcKRS-3 plasmid for generation of AcK95-RRM2. X.D. and G.C. wrote the paper. S.S.R., H.F., W.J.C., and P.W.D. edited the paper.

## Additional information

**Competing interests:** The authors declare no competing interests.

