## [Peer Review File · Nature Communications]

Reviewers' comments:

Reviewer #1 (Remarks to the Author):

In this manuscript, Dr. Xingming Deng and colleagues investigate the regulation of ribonucleotide reductase (RNR), the enzyme that catalyzes the rate limiting step in dNTP biosynthesis. RRM2 is one of the two RNR subunits and here the authors examine the regulation of RRM2 by acetylation. They identify an RRM2 acetylation site, K95, that is acetylated by KAT7 and deacetylated by SIRT2. Acetylation inhibits the activity of RNR and the authors suggest that this occurs by disruption of RRM2 homodimerization, a prerequisite for catalytic activity. The authors go on to show that RRM2 is deacetylated in S-phase and in response to DNA damage, conditions when elevated RNR activity is beneficial. Moreover, the authors provide evidence that RRM2 acetylation causes replication stress and suppresses tumorigenesis, likely through the inhibition RNR activity. Overall this is an interesting paper with a substantial amount of supporting data provides important new insights RNR regulation. However, there are a number of significant issues that should be addressed prior to publication, including the following:

1. The authors open the paper with data indicating that treatment of cells with deacetylase inhibitors inhibits RNR activity and lowers dNTP pool size. RNR regulation is tightly coupled to the cell cycle and therefore it is important to distinguish between direct control of RNR by deacetylases and secondary effects on RNR due to altered cell cycle dynamics. The authors should provide data detailing whether TSA + NAM treatment alters the cell cycle.
2. The signal to noise ratio in the EPR analyses (Fig 2B) is very low, suggesting that the recombinant protein has little to no Tyr radical. The results therefore do not provide sufficient evidence for the conclusion that the RRM2 mutants have equivalent Tyr radical content relative to the wild-type protein. For EPR analysis of recombinant protein, it also would be appropriate for the authors to quantitate spin by comparison with a standard.
3. On page 10, the authors summarize data concerning the expression of SIRT2 in cancer with the poorly word sentence: "Our findings suggest that Sirt2-reduced RRM2 acetylation could be a potential prognostic biomarker for NSCLC." First, it would be clearer to say '...SIRT2-regulated RRM2 acetylation status could be...' or 'RRM2 deacetylation by SIRT2 could be...'. More importantly, the conclusion is not fully supported by the data provided. The authors do not assess RRM2 acetylation status in the tumors in which SIRT2 expression was analyzed. SIRT2 has many substrates and the results do not establish that RRM2 targeting by SIRT2 is the key event in NSCLC.
4. The effects of SIRT2 inhibition on dNTP levels are reported to be statistically significant but are quite modest. What is the evidence that small changes in dNTP levels like this have a biologically significant impact? Also, since SIRT2 has many substrates as noted above, it is uncertain whether phenotypes associated with SIRT2 inhibition, such as suppression of cancer cell proliferation, are due to effects on dNTP levels. Can the effects of SIRT2 inhibition be reversed by the additional of exogenous nucleosides to cells?
5. Based on analysis of S150 RRM2 mutants, the authors conclude that IR- or CPT-induced RRM2 phosphorylation occurs exclusively at the S150 site. This conclusion is overstated, as the results indicate that RRM2 phosphorylation requires S150 but does not rule out contributions from other phosphorylation sites. There are many examples of priming phosphorylation events in which phosphorylation at one site promotes subsequent phosphorylation events on other sites.

6. Figure 4K shows SIRT2 staining of NSCLC samples. The authors should describe how the antibody used was validated to ensure that the signal does in fact reflect SIRT2 levels. In addition, the text refers to a lack of staining in adjacent normal tissues. This cannot be discerned from the images of tumor samples provided. There is a normal lung sample shown but it is not clear whether this is normal tissue from one of the same cases for which tumor staining is shown, or is from a normal, tumor-free individual.

Additional comments:

1. It is surprising that the authors do not mention that RRM2 is only one of two different proteins that can serve as the small beta subunit of the RNR holoenzyme. A second beta subunit, encoded by a separate gene (RRM2B), also can function in this manner. This is worth describing in the Introduction as well as considering further in the Discussion.

2. There is conflicting evidence in the literature regarding whether SIRT2 is an oncogene or tumor suppressor. Here the authors suggest that SIRT2 has tumor promoting activities. It would be appropriate for the authors to cite and discuss additional studies that either are in agreement with or run counter to their findings with SIRT2.

3. The authors use HPLC to measure dNTP levels in cells and provide a sample plot for dNTP standards (Fig S1). They should also include an example of the HPLC results for one of the actual experimental samples.

4. Since the arrest of cells by double thymidine block perturbs dNTP pools, it is not an ideal approach for assessing RNR regulation. It would be more appropriate to use an alternative approach, such as serum starvation in an appropriate cell line, to assess cell cycle regulation of RRM2 acetylation.

5. In a 2015 Molecular Cell paper, Buisson et al. define a mechanism whereby ATR signals through CHK1 and CDK to regulate RRM2 levels. This paper should be cited and discussed in the context of the new results concerning regulation of RRM2 by ATR reported by the authors.

6. The experiment concerning the ability of KAT7 to acetylate RRM2 (Fig 3B) in vitro doesn't appear to be included in the methods section. The reaction conditions and other details should be described.

Reviewer #2 (Remarks to the Author):

Review to the manuscript:

Acetylation dictates Ribonucleotide Reductase Function

by Chen et al.

The Deng lab presents an interesting functional study on the regulation of ribonucleotide reductase (RNR) function by post-translational lysine acetylation. The study shows many elegant in vivo and in vitro experiments, which were conducted in a technically sound way. Importantly, the authors evaluate a mechanism by which lysine acetylation controls RNR activity ultimately resulting in a decrease of RNR activity. Firstly, the authors observed that treatment of diverse lung cancer cell lines with trichostatin A (TSA) and nicotinamide (NAM), inhibitors for classical and sirtuin deacetylases, resulted in a decreased rate of C14-CDP conversion to dC14-CDP, as a measure for RNR activity. Next, the authors show that treatment with TSA and NAM deacetylase inhibitors results in an increase in

RRM2 acetylation. The authors were able to identify several lysine acetylation sites within the catalytic subdomain of RNR by mass spectrometry, namely RRM2. They show that K95 is the major acetyl-acceptor site, mutation of which completely abrogates RRM2 acetylation of ectopically expressed RRM2. Moreover, treatment of diverse human cell lines with TSA and NAM significantly reduced RNR activity compared to the non-treated controls. Using isotopically labeled C14-CDP Chen and co-workers analysed the conversion to dC14-CDP by RRM2 as a measure for RNR activity and examined the impact of corresponding K to R charge-conserving, non-acetylatable mutants of RRM2. In agreement for the importance of K95, they observed that K95R showed a high activity independent of the presence of TSA and NAM. The authors studied the mechanism underlying the reduction of RNR activity. RRM2 forms a dimer in solution. As K95 from one monomer forms a strong salt bridge towards E105 of a second monomer, they analysed the K to Q acetylation mimic mutant of RRM2 and observed that acetylation might interfere with homodimer formation. To this end, they performed immunoprecipitations following crosslinking and following SDS-PAGE. Additionally, they conducted analytical size-exclusion chromatography experiments of the purified proteins. Using siRNA libraries of 11 lysine-acetyltransferases (KATs) and the eleven classical deacetylases (HDAC1-11) and seven mammalian sirtuin deacetylases (Sirt1-7), they identified KAT7 as the RRM2 acetyltransferase and Sirt2 as RRM2 deacetylase. Analysis of the expression of Sirt2 and RRM2 revealed that both correlate during the cell cycle being high at the entry into S-phase supporting a functional and regulatory interaction. Chen et al. analysed the impact of RRM2 acetylation/deacetylation upon DNA damage, as this is reported to induce RNR activation. Performing kinase inhibition assays they observed, that pharmacological or siRNA-mediated inhibition of ATR kinase abolishes RRM2 phosphorylation at S150 resulting in an increase of RRM2 acetylation upon DNA damage. Conversely, phosphorylation of RRM2 by ATR upon DNA damage results in a decrease of RRM2 acetylation by increasing the interaction with the deacetylase Sirt2, leading to an increase in RNR activity. Finally, Chen and colleagues investigated the functional consequences of RRM2 acetylation in human lung cancer cells. Mutation of K95Q in RRM2 reduces replication fork speed, cell proliferation, colony forming capacity of H1299 human lung cancer cells and generation of lung cancer xenografts suggesting that RRM2 acetylation suppresses tumor growth. This work by the Deng lab comprises a highly interesting study for a huge research community. It is a further step towards understanding how proteins are regulated by lysine acetylation and shows that this post-translational can be a powerful to modify and regulate protein function. The study shows functional and mechanistic studies to show how acetylation at RRM2 results in the regulation of cellular processes. Furthermore, it shows that RRM2 acetylation is an important modification also in the physiological context. RRM2 acetylation and phosphorylation work in concert upon DNA damage to control the formation of dNTPs as appropriate. Using lung cancer cells and studies with xenografts, Chen and co-workers show that modulating RRM2 activity via its acetylation state might be a promising strategy also for therapeutic applications. As a summary, I highly recommend the publication of this manuscript in Nature Communications. However, I have some points shown below which should completely be addressed in a revised manuscript.

Minor Points:

Point 1:

Some figures are hardly to read (Fig 1d, Fig. 4d, Fig. 7d). Could you please enlarge these figures to be able to read the labels at the axes, the labels at the peaks in the MS spectra, the ion series in the MS spectra, the immunocytochemistry images, etc.

Point 2:

On page 14 you name K281Q. I guess you mean K283Q here.

Major points:

Point 1:

The experiments performed were all done using mutation of Lys to Gln as a mimic for lysine acetylation and Lys to Arg to mimic an non-acetylated state. However, as shown by several reports (of which none has been cited in this manuscript) these mutations are sometimes poor to mimic an acetylated/non-acetylated state at the molecular level. This has to be analysed on a site-specific basis. Gln is a good mimic if neutralisation of the positive charge at the lysine sidechain is the only effect of an acetylation event, while Arg can also mimic an acetylated state if lysine acetylation has also a steric contribution. Sometimes it is not possible to mimic an acetylation at all using Lys to Gln. Therefore, you should rephrase the sentence „replacement of lysine (K) with glutamine (Q) at a protein acetylation site is well established to mimic lysine acetylation“. Regarding the acetylation at K95 in RRM2, the salt bridge is clearly a strong one, as E105 and K95 are in an appropriate orientation to each other and in a distance of app. 2.6 Å. However, considering this distance, mutation of Lys to Arg at this site could also sterically interfere with RRM2 homodimer formation. This steric effect would also be realised upon acetylation of the lysine sidechain as an acetyl-lysine much bulkier compared to a Gln. As a summary, it has to be investigated if lysine acetylation also results in the dissociation of the RRM2 homodimer and if the Arg mutant is doing the same implicating steric effects of an acetylation at this site. The size-exclusion chromatography runs should be extended using the RRM2 K95R mutant. Furthermore, to confirm the mechanism that acetylation at K95 does result in dissociation of RRM2 homodimer formation, acetyl-lysine should be incorporated into RRM2 using a synthetic biological approach (genetic-code expansion concept). Several reports show the site-specific incorporation of acetyl-lysine into proteins (Knyphausen et al., 2016; de Boor et al, 2015). Apart from that, the size-exclusion chromatography runs (RRM2 K95R, K95Q, K95, and AcK95) should be performed using a calibrated S200 10/300 column or alternatively a Superose 6 10/300 column. The dimer peak shown for WT RRM2 on the S75 10/300 column seems to run in the void volume showing that the column is not suitable for correct determination of the oligomeric state.

Point 2:

In the results section (“Acetylation of RRM2 at K95 inactivates RNR”) you write that treatment of H1299 lung cancer cells with TSA and NAM results in a global increase in acetylation. As you only cited this, I am wondering if you really observed this experimentally. As you identified Sirt2 as RRM2 deacetylase it should be sufficient to use NAM as inhibitor. You should show how the acetylation level is altered upon treatment of cells with NAM, TSA individually and upon simultaneous treatment using both TSA and NAM. Moreover, you investigated the role of acetylation on the subdomain RRM2 of RNR. Did you analyse if also the regulatory subunit RRM1 is lysine acetylated?

Point 3:

For many experiments it is not obvious how you treated the cells. Do you always perform a treatment with deacetylase inhibitors TSA and NAM? This would stabilise acetylation events-if expressing the Lys to Gln or Lys to Arg mutant maybe at different positions apart from the ones being analysed. Can you please clarify for which experiments you applied the inhibitors.

As an example, in Fig. 1h you show the Arg mutants in TSA/NAM treated and untreated conditions, while for the Q mutants in Fig. 1j it is not obvious how you treated the cells. There are many more examples in this manuscript. Please state where you used the inhibitors and comment and explain how you decided when to use inhibitors and when not.

Point 4:

Related to point 3, it is not clear how you treated the cells for the mass-spectrometrical experiments. Measurements with and without treatment of NAM, TSA individually and in combination would show which sites are regulated by classical or sirtuin deacetylases. Please comment.

Point 5:

You examine the activity of RNR by analysing the conversion of C14-CDP to dC14-CDP. Is there any reason why using this NDP and not one of the others (TDP, GDP, ADP)?

Point 6:

The identification of deacetylases and acetyltransferases for RRM2 was done by applying siRNA. Please show the functionality of these siRNAs by showing knockdown of the enzymes by immunoblotting in the supplementary section.

Point 7:

The fluorescence microscopy figure (Fig. 4d) shows colocalisation of Sirt2 and RRM2 mostly in the cytosol. I assume that these cells are not synchronized and are mostly in different stages in interphase. As you show later, Sirt2 and RRM2 is mostly upregulated upon entry into S-phase. Please show the localisation of Sirt2 and RRM2 in cells in beginning S-phase to confirm that they also colocalise in particularly that cell cycle stage.

Point 8:

For some experiments you decided to analyse the Lys to Gln mutants and for others the Lys to Arg mutants. Please comment how you decided which to use.

Point 9:

For an loss-of-function modification as shown here for RRM2 acetylation at K95, it is necessary that acetylation has to occur at sufficiently high stoichiometries to result in an effect with physiological significance. To obtain a high stoichiometry, absence of deacetylase activity or presence of acetyltransferase activity might be important unless the acetylation occurs non-enzymatically. Please comment and discuss how RRM2 acetylation is regulated in the physiological context, even if not stress-induced, to reach these stoichiometries.

Reviewer #3 (Remarks to the Author):

In this manuscript, Xingming Deng and co-authors investigate regulation of Ribonucleotide Reductase (RNR) by acetylation/deacetylation. They propose that acetylation of the RRM2 subunit by KAT7 prevents its homodimerization thus inhibiting RNR activity, whereas deacetylation of RRM2 by SIRT2 allows RRM2 homodimerization, leading to activation of RNR and elevation of dNTP pools. Furthermore, the authors propose that upon DNA damage ATR activates RNR by phosphorylating RRM2 on S150, which promotes SIRT2 binding and deacetylation of RRM2. The proposed model is novel and interesting, but because several important experiments are lacking, at this stage it is not clear if this model is correct.

1. While it is convincingly shown in Figure 2 that K95Q mutation disrupts RRM2 dimerization, it is not proved that AcK95 also disrupts RRM2 dimerization. The dimerization status of the acetylated wild-type RRM2 should be investigated.

2. In Figures 3g and h it is shown that shRNA inhibition of KAT7 results in a significant downregulation of RRM2 acetylation and a concomitant ~1.6-fold increase in RNR activity. The authors should measure dNTP pools in the shRNA KAT7 cells to demonstrate that dNTP pools are increased (i.e. that RRM2 acetylation plays a role in keeping dNTP pools down).

3. Along the same lines, it should be investigated what fraction of RRM2 is acetylated before and after TSA/NAM treatment.

4. In Figure 5, the authors downregulate SIRT2 activity by Sirt2 siRNA or AGK2 and observe a decrease of dNTP pools. Parenthetically, it is not clear why some dNTPs decrease more than others and some do not decrease at all, and why the K95Q mutation in Figure 7b results in a different dNTP pool decrease pattern. However, more importantly, SIRT2 inhibition results in accumulation of cells in G1, as shown in Figure 5d. Because it is known that dNTP pools are low in G1, the observed decrease of dNTP pools might be an indirect effect of the accumulation of cells in G1. Furthermore, shouldn't we expect to see an accumulation of cells in S phase, and not in G1, if RNR activity is blocked by inactivation of SIRT2?

5. I am not sure if the title of the paper is very accurate ("Acetylation dictates ribonucleotide reductase function"). The function of RNR is to make dNTPs, and acetylation does not change it. The model proposes that acetylation inhibits ribonucleotide reductase activity.

Response to Reviewer #1:

Reviewers' comments:

Reviewer #1 (Remarks to the Author):

In this manuscript, Dr. Xingming Deng and colleagues investigate the regulation of ribonucleotide reductase (RNR), the enzyme that catalyzes the rate limiting step in dNTP biosynthesis. RRM2 is one of the two RNR subunits and here the authors examine the regulation of RRM2 by acetylation. They identify an RRM2 acetylation site, K95, that is acetylated by KAT7 and deacetylated by SIRT2. Acetylation inhibits the activity of RNR and the authors suggest that this occurs by disruption of RRM2 homodimerization, a prerequisite for catalytic activity. The authors go on to show that RRM2 is deacetylated in S-phase and in response to DNA damage, conditions when elevated RNR activity is beneficial. Moreover, the authors provide evidence that RRM2 acetylation causes replication stress and suppresses tumorigenesis, likely through the inhibition RNR activity. Overall this is an interesting paper with a substantial amount of supporting data provides important new insights RNR regulation. However, there are a number of significant issues that should be addressed prior to publication, including the following:

1. The authors open the paper with data indicating that treatment of cells with deacetylase inhibitors inhibits RNR activity and lowers dNTP pool size. RNR regulation is tightly coupled to the cell cycle and therefore it is important to distinguish between direct control of RNR by deacetylases and secondary effects on RNR due to altered cell cycle dynamics. The authors should provide data detailing whether TSA + NAM treatment alters the cell cycle.

Response: As suggested, cell cycle was analyzed following treatment of H1299 cells with TSA+NAM. Results reveal that TSA+NAM treatment did not significantly alter the cell cycle, indicating that the effects of TSA+NAM on RNR activity, dNTP pool size and DNA replication fork progression did not result from cell cycle change. These new data have been incorporated into new **Supplementary Fig.1e**, and described in revised text (**Page 5, lines 21-24**).

2. The signal to noise ratio in the EPR analyses (Fig 2B) is very low, suggesting that the recombinant protein has little to no Tyr radical. The results therefore do not provide sufficient evidence for the conclusion that the RRM2 mutants have equivalent Tyr radical content relative to the wild-type protein. For EPR analysis of recombinant protein, it also would be appropriate for the authors to quantitate spin by comparison with a standard.

Response: We agree that the signal to noise ratio in our previous EPR results (**old Fig. 2b**) was too low. As suggested, we repeated the EPR assay for analysis of the Tyr radical content in recombinant WT or mutant RRM2 proteins. New results reveal that the acetylation-mimetic K95Q and QQQQ proteins have EPR line shapes and peak-to-trough derivative amplitudes for the tyrosyl radical that are comparable to WT (**new Fig. 2b**). Double integration of the derivative-mode EPR spectra shows that the unpaired electron spin concentration is the same in each sample (**new Supplementary Fig. 2**). Therefore, the native tyrosyl radical site structure is not perturbed in the mutant proteins, and the

acetyl-mimetic RRM2 mutant protein is able to generate the same amount of tyrosyl radical cofactor as WT protein. These new data have been incorporated into new **Fig. 2b** and new **Supplementary Fig. 2**), and described in the revised text (**page 7, lines 17-23; page 8, line 1**).

3. On page 10, the authors summarize data concerning the expression of SIRT2 in cancer with the poorly worded sentence: “Our findings suggest that Sirt2-reduced RRM2 acetylation could be a potential prognostic biomarker for NSCLC.” First, it would be clearer to say ‘...SIRT2-regulated RRM2 acetylation status could be...’ or ‘RRM2 deacetylation by SIRT2 could be...’ More importantly, the conclusion is not fully supported by the data provided. The authors do not assess RRM2 acetylation status in the tumors in which SIRT2 expression was analyzed. SIRT2 has many substrates and the results do not establish that RRM2 targeting by SIRT2 is the key event in NSCLC.

Response: We currently do not have a specific acetyl-K95 (AcK95) antibody available for detecting RRM2 acetylation by IHC staining for NSCLC patient samples. We agree with the Reviewer’s comments since we did not directly assess RRM2 acetylation in the tumors due to the unavailability of specific acetyl-K95 (AcK95) antibody. According to our data, we changed the sentence to “Elevated levels of Sirt2 in tumor tissues were correlated with poor outcomes for NSCLC patients (**Supplementary Fig. 9**), suggesting that Sirt2, as the RRM2 deacetylase, could be a potential prognostic biomarker for NSCLC”. This description should more accurately reflect our Sirt2 IHC data from NSCLC patient samples. This change has been made in the revised text (**page 11, lines 15-18**).

4. The effects of SIRT2 inhibition on dNTP levels are reported to be statistically significant but are quite modest. What is the evidence that small changes in dNTP levels like this have a biologically significant impact? Also, since SIRT2 has many substrates as noted above, it is uncertain whether phenotypes associated with SIRT2 inhibition, such as suppression of cancer cell proliferation, are due to effects on dNTP levels. Can the effects of SIRT2 inhibition be reversed by the additional of exogenous nucleosides to cells?

Response: It is known that the dNTP pool size is tightly regulated in cells and an imbalance in the dNTP pool leads to disturbance of cell cycle progression, mutagenesis and apoptosis (*Science*. 345, 1512-5). Therefore, cells are very sensitive to any change in dNTP pool size, evidenced by a report that low dose of hydroxyurea (0.1 mM) moderately reduces dNTP levels (i.e. 20%) but fully blocks DNA replication (*J Biol Chem*. 261, 16037-42.). It has also been reported that a modest change in dTTP level (25%), caused by knockdown of Fhit, significantly hindered DNA replication and caused DNA double-strand breaks (*PLoS Genet*. 8, e1003077). Here we found that a modest reduction in dNTPs by inhibition of RNR activity induced by Sirt2 siRNA or Sirt2 inhibitor AGK2 led to suppression of cell proliferation and growth (new **Fig. 5d-h**), which supports and extends previous reports.

To test whether the effect of Sirt2 inhibition can be reversed by the addition of exogenous nucleosides to cells, as suggested, H1299 cells were transfected with Sirt2 siRNA followed by treatment with 20µM dNTPs for 24h as described (*J Cell Sci*. 114, 747-50; *Chromosoma* 108, 325-335). Cell proliferation was analyzed by BrdU incorporation as previously described (*PLoS one* 10, e0133786). Results show that treatment of cells with dNTPs reverses the inhibitory effect of Sirt2

siRNA on DNA synthesis and/or cell proliferation. These new data have been included as supplemental data **Supplementary Fig. 12**, and described in supplemental methods (**page 15, lines 8-11**) and in the revised text (**page 13, lines 11-15**).

5. Based on analysis of S150 RRM2 mutants, the authors conclude that IR- or CPT-induced RRM2 phosphorylation occurs exclusively at the S150 site. This conclusion is overstated, as the results indicate that RRM2 phosphorylation requires S150 but does not rule out contributions from other phosphorylation sites. There are many examples of priming phosphorylation events in which phosphorylation at one site promotes subsequent phosphorylation events on other sites.

Response: To avoid overstating the interpretation of Figure 6, as the reviewer suggested, we have revised the sentence to "...suggesting that IR- or CPT-induced RRM2 phosphorylation requires the S150 site" in the revised text (**page 15, lines 1-2**).

6. Figure 4K shows SIRT2 staining of NSCLC samples. The authors should describe how the antibody used was validated to ensure that the signal does in fact reflect SIRT2 levels. In addition, the text refers to a lack of staining in adjacent normal tissues. This cannot be discerned from the images of tumor samples provided. There is a normal lung sample shown but it is not clear whether this is normal tissue from one of the same cases for which tumor staining is shown, or is from a normal, tumor-free individual.

Response: To validate antibody specificity for Sirt2, first, endogenous Sirt2 was knocked down in H1299 cells using Sirt2 siRNA. Sirt2 expression in H1299 and H1299 Sirt2-silenced cells was analyzed by Western blot using Sirt2 specific antibody obtained from Santa Cruz Biotechnology (sc-28298). Results indicate that high levels of endogenous Sirt2 were observed in H1299 cells, and very low or undetectable Sirt2 levels were detected in Sirt2-silenced H1299 cells (**Supplementary Fig. 8a**), indicating endogenous Sirt2 was completely knocked down from parental H1299 cells. Second, immunofluorescence staining was performed using Sirt2 antibody in H1299 cells and H1299 Sirt2-silenced cells. Results show that H1299 cells had a strong Sirt2 (red) signal while H1299 Sirt2-silenced cells had no detectable Sirt2 observed (**Supplementary Fig. 8b**), consistent with our Western blot data. These results confirm that the Sirt2 antibody used in these experiments is specific for the detection of Sirt2. These new data have been included in **Supplementary Fig. 8**, and described in the revised text (**page 10, lines 15-16**).

Regarding the previous Figure 4K, we have added new data to compare Sirt2 expression in normal lung tissues versus lung cancer tissues for each representative case. Normal tissues are the adjacent normal lung tissues from the same cases. These new data have been included in **Supplementary Fig. 9**, and described in the revised text (**page 11, lines 5-14**).

Additional comments:

1. It is surprising that the authors do not mention that RRM2 is only one of two different proteins that can serve as the small beta subunit of the RNR holoenzyme. A second beta subunit, encoded by a separate gene (RRM2B), also can function in this manner. This is worth describing in the Introduction

as well as considering further in the Discussion.

Response: RRM2B, also called p53R2, is a second radical-providing small subunit in mammalian cells. p53R2 is induced by p53 and involved in regulating DNA replication and repair, cell cycle arrest, and mitochondrial homeostasis. As suggested, we have included a description of RRM2B (p53R2) in the “Introduction” in the revised text (**page 3, lines 14-17**).

2. *There is conflicting evidence in the literature regarding whether SIRT2 is an oncogene or tumor suppressor. Here the authors suggest that SIRT2 has tumor promoting activities. It would be appropriate for the authors to cite and discuss additional studies that either are in agreement with or run counter to their findings with SIRT2.*

Response: As requested, we cited and added additional discussion regarding Sirt2’s tumor suppressor and oncogenic functions as: “Interestingly, Sirt2 has been reported to have both tumor suppressor (Kim, H.S. *et al. Cancer cell* **20**, 487-499; Hiratsuka, M. *et al. Biochemical and biophysical research communications* **309**, 558-566; Serrano, L. *et al. Genes & development* **27**, 639-653; Serrano, L. *et al. J. Biol. Chem* **292**, 9919-9931) and oncogenic functions (Grbesa, I. *et al. PLoS One* **10**, e0124670; Liu, P.Y. *et al. Cell Death Differ* **20**, 503-514; Chen, J. *et al. Hepatology* **57**, 2287-2298; McGlynn, L.M. *et al. European journal of cancer* **50**, 290-301). On one hand, Sirt2-deficient animals exhibit genomic instability and chromosomal aberrations and are prone to tumorigenesis (Serrano, L. *et al. Genes & development* **27**, 639-653). Sirt2-knockout mice develop gender-specific tumorigenesis, with females primarily developing mammary tumors, and males more often developing hepatocellular carcinoma (HCC) (Kim, H.S. *et al. Cancer cell* **20**, 487-499). Sirt2 functions as a tumor suppressor through its role in regulating mitosis and genome integrity. On the other hand, Sirt2 has also been reported to have tumor-promoting activity (Grbesa, I. *et al. PLoS One* **10**, e0124670; Liu, P.Y. *et al. Cell Death Differ* **20**, 503-514; Chen, J. *et al. Hepatology* **57**, 2287-2298; McGlynn, L.M. *et al. European journal of cancer* **50**, 290-301) . Sirt2 not only stabilizes Myc oncoproteins to enhance the proliferation of various cancer cell types (Liu, P.Y. *et al. Cell Death Differ* **20**, 503-514) but also promotes motility and invasiveness of HCC cells by regulating the PKB/GSK-3 β / β -catenin signaling pathway (Chen, J. *et al. Hepatology* **57**, 2287-2298). Our and others’ findings demonstrated that higher levels of Sirt2 in NSCLC patients are associated with poor prognosis (Grbesa, I. *et al. PLoS One* **10**, e0124670) (**Supplementary Fig. 9**). Moreover, inhibition of Sirt2 by selective Sirt2 inhibitor (thiomristoyl lysine compound) exhibits broad anticancer activity (Jing, H. *et al. Cancer Cell* **29**, 297- 310). In support of Sirt2’s oncogenic function, here we discovered that Sirt2 deacetylates RRM2 to enhance RNR activity leading to increased dNTPs pool size and proliferation of cancer cells.”
These additional discussions have been included in the revised text (**page 18, lines 7-21**).

3. *The authors use HPLC to measure dNTP levels in cells and provide a sample plot for dNTP standards (Fig S1). They should also include an example of the HPLC results for one of the actual experimental samples.*

Response: As suggested, HPLC sample plots of dNTPs for one of the actual experiments following treatment of H1299 cells with a combination of TSA and NAM have been incorporated into **Supplementary Fig. 1c, left panel**).

4. *Since the arrest of cells by double thymidine block perturbs dNTP pools, it is not an ideal approach for assessing RNR regulation. It would be more appropriate to use an alternative approach, such as serum starvation in an appropriate cell line, to assess cell cycle regulation of RRM2 acetylation.*

Response: As suggested, in addition to double thymidine block, we also used serum starvation as an alternative approach to assess cell cycle regulation of RRM2 acetylation. H1299 cells were synchronized at the G0/G1 stage by serum starvation, followed by re-addition of serum (10% FBS) to allow cells to re-enter the cell cycle. Cells entered S phase at 6 h after serum re-addition. Similarly, significant increased levels of Sirt2 (**Supplementary Fig. 11a**) in association with decreased levels of RRM2 acetylation (**Supplementary Fig. 11b**) were observed in S/G2 phase (6-28 h). These results provide further evidence that Sirt2 regulation of RRM2 deacetylation occurs in a cell cycle dependent manner. These new data have been included as **Supplementary Fig.11**, and described in the revised text (**page 12, lines 9-15**).

5. *In a 2015 Molecular Cell paper, Buisson et al. define a mechanism whereby ATR signals through CHK1 and CDK to regulate RRM2 levels. This paper should be cited and discussed in the context of the new results concerning regulation of RRM2 by ATR reported by the authors.*

Response: It has recently been reported that the ATR/Chk1 pathway plays a key role in promoting RRM2 accumulation by stabilizing E2F1 in S phase cells, which may be important for countering the replication stress in cancer cells (Buisson et al, *Mol Cell* **59**, 1011-1024). As suggested, we cited and discussed this important paper in the revised text (**page 18, lines 23-24; page 19, lines 1-2**).

6. *The experiment concerning the ability of KAT7 to acetylate RRM2 (Fig 3B) in vitro doesn't appear to be included in the methods section. The reaction conditions and other details should be described.*

Response: As suggested, the methods section relevant for the study of the direct acetylation of RRM2 in vitro by KAT7 for new Figure 3f has been included in "Supplemental Methods" (**page 16, lines 18-26**).

Response to Reviewer #2: Reviewer

#2 (Remarks to the Author): Review to

the manuscript:

Acetylation dictates Ribonucleotide Reductase Function

by Chen et al.

The Deng lab presents an interesting functional study on the regulation of ribonucleotide reductase (RNR) function by post-translational lysine acetylation. The study shows many elegant in vivo and in

vitro experiments, which were conducted in a technically sound way. Importantly, the authors evaluate a mechanism by which lysine acetylation controls RNR activity ultimately resulting in a decrease of RNR activity. Firstly, the authors observed that treatment of diverse lung cancer cell lines with trichostatin A (TSA) and nicotinamide (NAM), inhibitors for classical and sirtuin deacetylases, resulted in a decreased rate of C14-CDP conversion to dC14-CDP, as a measure for RNR activity. Next, the authors show that treatment with TSA and NAM deacetylase inhibitors results in an increase in RRM2 acetylation. The authors were able to identify several lysine acetylation sites within the catalytic subdomain of RNR by mass spectrometry, namely RRM2. They show that K95 is the major acetyl-acceptor site, mutation of which completely abrogates RRM2 acetylation of ectopically expressed RRM2. Moreover, treatment of diverse human cell lines with TSA and NAM significantly reduced RNR activity compared to the non-treated controls. Using isotopically labeled C14-CDP Chen and co-workers analysed the conversion to dC14-CDP by RRM2 as a measure for RNR activity and examined the impact of corresponding K to R charge-conserving, non-acetylatable mutants of RRM2. In agreement for the importance of K95, they observed that K95R showed a high activity independent of the presence of TSA and NAM. The authors studied the mechanism underlying the reduction of RNR activity. RRM2 forms a dimer in solution. As K95 from one monomer forms a strong salt bridge towards E105 of a second monomer, they analysed the K to Q acetylation mimic mutant of RRM2 and observed that acetylation might interfere with homodimer formation. To this end, they performed immunoprecipitations following crosslinking and following SDS-PAGE. Additionally, they conducted analytical size-exclusion chromatography experiments of the purified proteins. Using siRNA libraries of 11 lysine-acetyltransferases (KATs) and the eleven classical deacetylases (HDAC1-11) and seven mammalian sirtuin deacetylases (Sirt1-7), they identified KAT7 as the RRM2 acetyltransferase and Sirt2 as RRM2 deacetylase. Analysis of the expression of Sirt2 and RRM2 revealed that both correlate during the cell cycle being high at the entry into S-phase supporting a functional and regulatory interaction. Chen et al. analysed the impact of RRM2 acetylation/deacetylation upon DNA damage, as this is reported to induce RNR activation. Performing kinase inhibition assays they observed, that pharmacological or siRNA-mediated inhibition of ATR kinase abolishes RRM2 phosphorylation at S150 resulting in an increase of RRM2 acetylation upon DNA damage. Conversely, phosphorylation of RRM2 by ATR upon DNA damage results in a decrease of RRM2 acetylation by increasing the interaction with the deacetylase Sirt2, leading to a increase in RNR activity. Finally, Chen and colleagues investigated the functional consequences of RRM2 acetylation in human lung cancer cells. Mutation of K95Q in RRM2 reduces replication fork speed, cell proliferation, colony forming capacity of H1299 human lung cancer cells and generation of lung cancer xenografts suggesting that RRM2 acetylation suppresses tumor growth. This work by the Deng lab comprises a highly interesting study for a huge research community. It is a further step towards understanding how proteins are regulated by lysine acetylation and shows that this post-translational can be a powerful to modify and regulate protein function. The study shows functional and mechanistic studies to show how acetylation at RRM2 results in the regulation of cellular processes. Furthermore, it shows that RRM2 acetylation is an important modification also in the physiological context. RRM2 acetylation and phosphorylation work in concert upon DNA damage to control the formation of dNTPs as appropriate. Using lung cancer cells and studies with xenografts, Chen and co-workers show that modulating RRM2 activity via its acetylation state might be a promising strategy also for therapeutic applications. As a summary, I highly recommend the publication of this manuscript in Nature Communications. However, I have some points shown below which should completely be addressed in a revised

manuscript.

Minor Points:

Point 1:

Some figures are hardly to read (Fig 1d, Fig. 4d, Fig. 7d). Could you please enlarge these figures to be able to read the labels at the axes, the labels at the peaks in the MS spectra, the ion series in the MS spectra, the immunocytochemistry images, etc.

Response: As suggested, Fig 1d, Fig. 4d and Fig. 7d have been enlarged.

Point 2:

On page 14 you name K281Q. I guess you mean K283Q here.

Response: Yes, this should be K283Q. We have corrected this in the revised text (new **page 16, line 8**).

Major points:

Point 1:

The experiments performed were all done using mutation of Lys to Gln as a mimic for lysine acetylation and Lys to Arg to mimic a non-acetylated state. However, as shown by several reports (of which none has been cited in this manuscript) these mutations are sometimes poor to mimic an acetylated/non-acetylated state at the molecular level. This has to be analysed on a site-specific basis. Gln is a good mimic if neutralisation of the positive charge at the lysine sidechain is the only effect of an acetylation event, while Arg can also mimic an acetylated state if lysine acetylation has also a steric contribution. Sometimes it is not possible to mimic an acetylation at all using Lys to Gln. Therefore, you should rephrase the sentence „replacement of lysine (K) with glutamine (Q) at a protein acetylation site is well established to mimic lysine acetylation“. Regarding the acetylation at K95 in RRM2, the salt bridge is clearly a strong one, as E105 and K95 are in an appropriate orientation to each other and in a distance of app. 2.6 Å. However, considering this distance, mutation of Lys to Arg at this site could also sterically interfere with RRM2 homodimer formation. This steric effect would also be realised upon acetylation of the lysine sidechain as an acetyl-lysine much bulkier compared to a Gln. As a summary, it has to be investigated if lysine acetylation also results in the dissociation of the RRM2 homodimer and if the Arg mutant is doing the same implicating steric effects of an acetylation at this site. The size-exclusion chromatography runs should be extended using the RRM2 K95R mutant. Furthermore, to confirm the mechanism that acetylation at K95 does result in dissociation of RRM2 homodimer formation, acetyl-lysine should be incorporated into RRM2 using a synthetic biological approach (genetic-code expansion concept). Several reports show the site-specific incorporation of acetyl-lysine into proteins (Knyphausen et al., 2016; de Boer et al, 2015). Apart from that, the size-exclusion chromatography runs (RRM2 K95R, K95Q, K95, and AcK95) should be performed using a calibrated

S200 10/300 column or alternatively a Superose 6 10/300 column. The dimer peak shown for WT RRM2 on the S75 10/300 column seems to run in the void volume showing that the column is not suitable for correct determination of the oligomeric state.

Response: As suggested, we rephrased the sentence "replacement of lysine (K) with glutamine (Q) at a protein acetylation site is well established to mimic lysine acetylation " to "replacement of lysine (K) with glutamine (Q) at a protein acetylation site has been reported to mimic lysine acetylation when the function of the acetylation event neutralizes the positive charge of the lysine side chain "(Inuzuka et al, *Cell* 2012, **150**: 179-193; *Proc Natl Acad Sci U S A.* 2015 ;112:E3679-88; *J Biol Chem.* 2016, 291:14677-94). In the case of RRM2 K95 acetylation, K95 in RRM2 forms a strong salt bridge with E105 and K95 through its positive charge side chain. Thus, the K95Q mutation is able to mimic RRM2 acetylation. This change has been made in the revised text (**page 7, lines 4-6; page 17, lines 11-13**).

To directly evaluate the role of actual acetylation at K95 in regulating RRM2 homodimerization in vitro, as suggested, in addition to acetyl-mimetic mutant RRM2 K95Q, we employed a genetic-code expansion concept to specifically incorporate acetyl-lysine into RRM2 at the K95 site as recently described (Neumann H et al, *Nat Chem Biol* **4**, 232-234; de Boor and Lammers et al, *Proc Natl Acad Sci U S A.* 2015, 112:E3679-88; Knyphausen and Lammers et al., *J Biol Chem.* 2016, 291:14677-94). We obtained the pRSF-Duet1/MbtRNA_{CUA}/AcKRS-3 construct from Dr. Michael Lammers's laboratory (University of Cologne, Germany). The construct contains the coding regions for acetyl-lysyl-tRNA-synthetase (AcKRS3) and the cognate amber suppressor MbtRNA_{CUA} from *Methanosarcina barkeri* (**Supplementary Fig. 3a**). First, human RRM2 cDNA was cloned into the pRSF-Duet1-MbtRNA_{CUA}/AcKRS-3 between the BamH I and Not I sites. Then, the QuikChange site-directed mutagenesis kit (Agilent Technologies, Santa Clara, CA) was used to mutate the code of K95 (AAG) to TAG, which can exclusively be recognized by MbtRNA_{CUA} for acetyl-lysine incorporation. The resulting acetyl K95 RRM2 (AcK95-RRM2) constructs were transformed into *E.coli* BL21(DE3) to produce recombinant AcK95 protein with supplementation of N_ε-Acetyl-L-lysine (AcK) in LB medium (**Supplementary Fig. 3b**). After the production of recombinant RRM2 WT (K95), K95Q, AcK95 and K95R proteins, the size-exclusion chromatography (SEC) experiments were performed as described in "Supplemental Methods". New results indicate that the majority of WT and K95R proteins displayed a homodimer state. In contrast, most K95Q and AcK95 RRM2 molecules were detected as monomers. These findings reveal that acetylation of RRM2 at K95 (AcK95) or K95 acetyl-mimetic mutation of RRM2 (K95Q) suppresses RRM2 homodimerization (new **Fig. 2f**). Moreover, purified WT, K95Q, AcK95 and K95R proteins were cross-linked with disuccinimidylsuberate (DSS). Consistent with the SEC data, WT and K95R RRM2 proteins were mainly detected as a dimer, while K95Q and AcK95 RRM2 proteins were mainly detected as a monomer (new **Fig. 2g**). Collectively, these results suggest that acetylation of RRM2 at K95 interferes with the ability of RRM2 molecules to form functional homodimers. These new data have been incorporated into **Fig. 2f-g**, and described in the revised text (**page 8, lines 13-24; page 9, lines 1-9**).

The "Method" for site-specific incorporation of N_ε-Acetyl-L-lysine into RRM2 protein was included in the revised text (**page 21, lines 20-24; page 22, lines 1-19**).

Point 2:

In the results section (“Acetylation of RRM2 at K95 inactivates RNR”) you write that treatment of H1299 lung cancer cells with TSA and NAM results in a global increase in acetylation. As you only cited this, I am wondering if you really observed this experimentally. As you identified Sirt2 as RRM2 deacetylase it should be sufficient to use NAM as inhibitor. You should show how the acetylation level is altered upon treatment of cells with NAM, TSA individually and upon simultaneous treatment using both TSA and NAM. Moreover, you investigated the role of acetylation on the subdomain RRM2 of RNR. Did you analyse if also the regulatory subunit RRM1 is lysine acetylated?

Response: As suggested, we detected global acetylation experimentally. H1299 cells were treated with TSA (2 μ M), NAM (10mM) or in combination for 18h, followed by Western blot using pan-acetyl lysine antibody for measurement of global acetylation. Results indicate that TSA or NAM enhanced global acetylation of multiple proteins in different panels but there are many overlaps. The combination of TSA with NAM had additive effects on global acetylation. These new data have been included as **Supplementary Fig. 1a**, and described in the revised text (**page 5, lines 5-9**).

As the reviewer mentioned, we also analyzed RRM1 acetylation. We observed detectable RRM1 acetylation by Western blot, but could not identify the acetylation site using LC/MS analysis (we performed LC/MS several times). We are focused on RRM2 acetylation in this manuscript. Studying RRM1 acetylation is an interesting topic that deserves further investigation in future studies.

Point 3:

For many experiments it is not obvious how you treated the cells. Do you always perform a treatment with deacetylase inhibitors TSA and NAM? This would stabilise acetylation events-if expressing the Lys to Gln or Lys to Arg mutant maybe at different positions apart from the ones being analysed. Can you please clarify for which experiments you applied the inhibitors.

As an example, in Fig. 1h you show the Arg mutants in TSA/NAM treated and untreated conditions, while for the Q mutants in Fig. 1j it is not obvious how you treated the cells. There are many more examples in this manuscript. Please state where you used the inhibitors and comment and explain how you decided when to use inhibitors and when not.

Response: As the reviewer suggested, if we used TSA/NAM in Figures, we labeled TSA/NAM in these Figures and described in figure legends. If we did not use TSA/NAM in Figures, we did not label TSA/NAM in those Figures and did not describe in their figure legends. For example, Figure 1a,b, c, d, e, g, h used TSA/NAM as labelled and described in figure legend. For Figure 1i and 1j, we did not use TSA/NAM, therefore, we did not label TSA/NAM in Figure 1i and 1j, and did not state TSA/NAM treatment in the legend of Figure 1i and 1J.

We did not use TSA/NAM in Figures 2, 3, 4, 5, 6 and 7 and accordingly did not label TSA/NAM in Figures 2, 3, 4, 5, 6 and 7.

A brief explanation of the reasons to use TSA/NAM in Figure 1: for Figure 1a-b, we used TSA/NAM to test whether treatment with deacetylase inhibitors affects intracellular RNR activity. Figure 1c tested whether RRM2 is regulated by acetylation, and whether inhibition of deacetylase by TSA/NAM enhances RRM2 acetylation levels in various cells. For Figure 1d, please see response to “Point 4”. To identify which site(s) is the major acetylation site(s) to regulate RNR activity, we mutated Flag-

RRM2 at each of the individual acetylation sites, or at all four sites simultaneously, from lysine (K) to arginine (R) to eliminate acetylation, and generated K30R, K61R, K95R, K283R and the compound K30R/K61R/K95R/K283R (RRRR) RRM2 mutants as shown in Figure 1e, g and h. We used TSA/NAM to test whether inhibition of deacetylases affects the acetylation and RNR enzymatic activity of WT RRM2 or K30R, K61R, K95R, K283R and RRRR mutant RRM2. Results indicate K95 is the major acetylation site to regulate RNR activity. Please see detailed descriptions in the revised text (pages 5, 6 and 7).

Point 4:

Related to point 3, it is not clear how you treated the cells for the mass-spectrometrical experiments. Measurements with and without treatment of NAM, TSA individually and in combination would show which sites are regulated by classical or sirtuin deacetylases. Please comment.

Response: The original intention for the treatment of cells with NAM+TSA was to obtain a stronger signal of acetylation for LC/MS analysis to identify possible RRM2 acetylation site(s), because the identification of protein acetylation site(s) presents a technical challenge. In **Fig.1d**, mass-spectrometrical (LC/MS) analysis was performed following treatment of cells with the combination of NAM and TSA. As the reviewer suggested, we also performed the LC/MS analysis following treatment with individual NAM, TSA or no treatment control compared to the combination of NAM+TSA. Intriguingly, the acetylation sites were the same (K30, K61, K95 and K283) following various treatments, indicating that NAM+TSA only enhances RRM2 acetylation levels as shown in **Fig.1c**, but does not affect acetylation sites in RRM2. RRM2 has four acetylation sites (K30, K61, K95 and K283) that could be regulated by acetyltransferase and deacetylase in physical cell growth conditions. This is understandable because acetylation sites are dependent on RRM2 itself and regulated by endogenous acetyltransferase deacetylase in cells. NAM/TSA only affects RRM2 acetylation level. Based on our findings, among these four sites, K95 is the major site. The LC/MS technology is used for identifying protein acetylation site(s) and not for characterizing signal pathways. We have discovered that RRM2 is directly acetylated by KAT7 (**Fig. 3**), and deacetylated by Sirt2 (**Fig.4**), respectively.

Point 5:

You examine the activity of RNR by analysing the conversion of C14-CDP to dC14-CDP. Is there any reason why using this NDP and not one of the others (TDP, GDP, ADP)?

Response: Measurement of RNR activity by analyzing the conversion of C14-CDP to dC14-CDP is the conventional method. It has been shown that the K_m for RNR catalyzing CDP is $3.2 \pm 0.55 \mu\text{M}$, K_m for RNR catalyzing GDP is $500 \pm 84 \mu\text{M}$, K_m for RNA catalyzing ADP is $480 \pm 97 \mu\text{M}$, K_m for RNR catalyzing UDP is $72 \pm 15 \mu\text{M}$ (*J Biol Chem.* 273, 34098-104), indicating that CDP is most sensitive substrate for measuring RNR activity. This is why we used C14-CDP as substrate to measure RNR activity.

Point 6:

The identification of deacetylases and acetyltransferases for RRM2 was done by applying siRNA. Please show the functionality of these siRNAs by showing knockdown of the enzymes by

immunoblotting in the supplementary section.

Response: As suggested, we have provided immunoblotting (Western blot) evidence for silencing of various acetyltransferases in **Supplementary Fig. 4**, and immunoblotting evidence for silencing of various deacetylases in **Supplementary Fig. 5**. These new data have been included and described in the revised text (**page 9, lines 13-16; page 10, lines 5-7**).

Point 7:

The fluorescence microscopy figure (Fig. 4d) shows colocalisation of Sirt2 and RRM2 mostly in the cytosol. I assume that these cells are not synchronized and are mostly in different stages in interphase. As you show later, Sirt2 and RRM2 is mostly upregulated upon entry into S-phase. Please show the localisation of Sirt2 and RRM2 in cells in beginning S-phase to confirm that they also colocalise in particularly that cell cycle stage.

Response: To determine the co-localization of Sirt2 and RRM2 in G1 and S phase cells, as suggested, S phase or G1 phase H1299 cells were collected at 2h or 14h time points following double thymidine block release. Co-immunostaining experiments were performed using RRM2 and Sirt2 antibodies in G1 and S cells. Significantly increased co-localization of Sirt2 and RRM2 was observed in S phase cells compared to G1 phase cells. These new data have been included as **Supplementary Fig. 7**, and described in the revised text (**page 10, lines 13-16**).

Point 8:

For some experiments you decided to analyse the Lys to Gln mutants and for others the Lys to Arg mutants. Please comment how you decided which to use.

Response: Lys (K) to Gln (Q) mutants were used to mimic lysine acetylation when the function of the acetylation event neutralizes the positive charge of the lysine side chain as we described in response to “*Point 1*” in “*Major points*” section. In contrast, Lys (K) to Arg (R) mutants were used to eliminate acetylation leading to the protein becoming acetylation-deficient at the mutated site. If we need to evaluate the function or activity of protein acetylation, we will use Lys to Gln mutants. If we need to evaluate the function or activity of acetylation-deficient protein, we will use Lys to Arg mutants.

Point 9:

For an loss-of-function modification as shown here for RRM2 acetylation at K95, it is necessary that acetylation has to occur at sufficiently high stoichiometries to result in an effect with physiological significance. To obtain a high stoichiometry, absence of deacetylase activity or presence of acetyltransferase activity might be important unless the acetylation occurs non-enzymatically. Please comment and discuss how RRM2 acetylation is regulated in the physiological context, even if not stress-induced, to reach these stoichiometries.

Response: Our data show that the RRM2 deacetylase Sirt2 was expressed at very low level in G1 phase (14h and 16h time points after double-thymidine block, **Fig.5a**). This low expression of deacetylase Sirt2 in G1 phase could contribute to increased RRM2 acetylation to sufficiently high stoichiometry (**Fig. 5b, c**, left sides, at 14h and 16h time points) that can ensure physiologically

required low level of RNR activity in G1 phase. When cells enter S phase (2-6h after double-thymidine block), Sirt2 is significantly increased (**Fig.5a**), which led to reduction of RRM2 acetylation (**Fig. 5b, c**, left sides, at 2h and 6h time points) to ensure RNR activity above a physiological threshold level required for dNTPs synthesis in S phase. Thus, the dynamic regulation of RNR activity by Sirt2-mediated RRM2 deacetylation may contribute to the physiological regulation of cell cycle progression and cell growth.

Response to Reviewer #3:

Reviewer #3 (Remarks to the Author):

In this manuscript, Xingming Deng and co-authors investigate regulation of Ribonucleotide Reductase (RNR) by acetylation/deacetylation. They propose that acetylation of the RRM2 subunit by KAT7 prevents its homodimerization thus inhibiting RNR activity, whereas deacetylation of RRM2 by SIRT2 allows RRM2 homodimerization, leading to activation of RNR and elevation of dNTP pools. Furthermore, the authors propose that upon DNA damage ATR activates RNR by phosphorylating RRM2 on S150, which promotes SIRT2 binding and deacetylation of RRM2. The proposed model is novel and interesting, but because several important experiments are lacking, at this stage it is not clear if this model is correct.

1. While it is convincingly shown in Figure 2 that K95Q mutation disrupts RRM2 dimerization, it is not proved that AcK95 also disrupts RRM2 dimerization. The dimerization status of the acetylated wild-type RRM2 should be investigated.

Response: To directly evaluate the role of actual acetylation at K95 in regulating RRM2 homodimerization in vitro, as suggested, in addition to acetyl-mimetic mutant RRM2 K95Q, we employed a genetic-code expansion concept to specifically incorporate acetyl-lysine into RRM2 at the K95 site as recently described (Neumann H et al, *Nat Chem Biol* **4**, 232-234; de Boor and Lammers et al, *Proc Natl Acad Sci U S A*. 2015, 112:E3679-88; Knyphausen and Lammers et al., *J Biol Chem*. 2016, 291:14677-94). We obtained the pRSF-Duet1/MbtRNA_{CUA}/AcKRS-3 construct from Dr. Michael Lammers's laboratory (University of Cologne, Germany). The construct contains the coding regions for acetyl-lysyl-tRNA-synthetase (AcKRS3) and the cognate amber suppressor MbtRNA_{CUA} from *Methanosarcina barkeri* (**Supplementary Fig. 3a**). First, human RRM2 cDNA was cloned into the pRSF-Duet1-MbtRNA_{CUA}/AcKRS-3 between the BamH I and Not I sites. Then, the QuikChange site-directed mutagenesis kit (Agilent Technologies, Santa Clara, CA) was used to mutate the code of K95 (AAG) to TAG, which can exclusively be recognized by MbtRNA_{CUA} for acetyl-lysine incorporation. The resulting acetyl K95 RRM2 (AcK95-RRM2) constructs were transformed into *E.coli* BL21(DE3) to produce recombinant AcK95 protein with supplementation of N_ε-Acetyl-L-lysine (AcK) in LB medium (**Supplementary Fig. 3b**). After the production of recombinant RRM2 WT (K95), K95Q, AcK95 and K95R proteins, size-exclusion chromatography (SEC) experiments were performed as described in "Supplemental Methods". New results indicate that the majority of WT and K95R proteins displayed a homodimer state. In contrast, most K95Q and AcK95 RRM2 molecules were detected as monomers. These findings reveal that acetylation of RRM2 at K95 (AcK95) or K95 acetyl-mimetic mutation of RRM2 (K95Q) suppresses RRM2 homodimerization (new **Fig. 2f**). Moreover, purified WT, K95Q,

AcK95 and K95R proteins were cross-linked with disuccinimidylsuberate (DSS). Consistent with the SEC data, WT and K95R RRM2 proteins were mainly detected as a dimer, while K95Q and AcK95 RRM2 proteins were mainly detected as a monomer (new **Fig. 2g**). Collectively, these results suggest that acetylation of RRM2 at K95 interferes with the ability of RRM2 molecules to form functional homodimers. These new data have been incorporated into **Fig. 2f-g**, and described in the revised text (**page 8, lines 13-24; page 9, lines 1-9**).

The “Method” for site-specific incorporation of N_ε-Acetyl-L-lysine into RRM2 protein was included in the revised text (**page 21, lines 20-24; page 22, lines 1-19**).

2. In Figures 3g and h it is shown that shRNA inhibition of KAT7 results in a significant downregulation of RRM2 acetylation and a concomitant ~1.6-fold increase in RNR activity. The authors should measure dNTP pools in the shRNA KAT7 cells to demonstrate that dNTP pools are increased (i.e. that RRM2 acetylation plays a role in keeping dNTP pools down).

Response: For Figure 3, as suggested, we measured dNTP levels in H1299 cells expressing KAT7 shRNA1, KAT7 shRNA2 or Ctrl shRNA. Results indicate that knockdown of KAT7 resulted in elevated levels of dNTPs. These new data have been incorporated into revised Figure 3 as **Figure 3j**, described in the revised text (**page 9, line 22-23; page 10, line 1**).

3. Along the same lines, it should be investigated what fraction of RRM2 is acetylated before and after TSA/NAM treatment.

Response: As suggested, we compared acetylation sites of RRM2 before and after TSA/NAM treatment. The acetylation sites were the same (K30, K61, K95 and K283) before and after TSA/NAM treatment, indicating that NAM+TSA only enhances RRM2 acetylation levels, but does not change the acetylation sites in RRM2. RRM2 has four acetylation sites (K30, K61, K95 and K283) that could be regulated by acetyltransferase and deacetylase in physical cell growth conditions. This is understandable because acetylation sites are dependent on RRM2 itself and regulated by endogenous acetyltransferase and deacetylase in cells. According to our data, K95 is the major acetylation site of RRM2 among these four acetylation sites.

4. In Figure 5, the authors downregulate SIRT2 activity by Sirt2 siRNA or AGK2 and observe a decrease of dNTP pools. Parenthetically, it is not clear why some dNTPs decrease more than others and some do not decrease at all, and why the K95Q mutation in Figure 7b results in a different dNTP pool decrease pattern. However, more importantly, SIRT2 inhibition results in accumulation of cells in G1, as shown in Figure 5d. Because it is known that dNTP pools are low in G1, the observed decrease of dNTP pools might be an indirect effect of the accumulation of cells in G1. Furthermore, shouldn't we expect to see an accumulation of cells in S phase, and not in G1, if RNR activity is blocked by inactivation of SIRT2?

Response: It has been reported that inhibition of RNR by hydroxyurea (3mM) induces decrease of dATP, dCTP and dGTP, but there is no significant effect on dTTP level (*J Biol Chem.* 1986 Dec 5;

261(34):16037-42.). Additionally, a recent study showed that depletion of IRBIT, an allosteric inhibitor of RNR, caused upregulation of dATP, dCTP and dGTP, but there was no significant effect on dTTP level, too (*Science*. 2014 Sep 19;345(6203):1512-5.). Here we found that inhibition of RRM2 deacetylase Sirt2 resulted in significant decrease in both RNR activity (**Fig.5d, e**) and dNTPs pool size (**Fig.5f**). However, the degree of dTTP reduction was less than that of the other three types of dNTPs, i.e. dATP, dCTP and dGTP (**Fig.5f**). Our findings have some similarity with previous reports. Our findings and others' suggest that RNR may have slightly less regulatory effect on dTTP than the other three dNTPs, but the mechanism is currently unclear. It is possible that, in addition to RNR, the dTTP pool size can also be regulated by thymidine kinase 1 (TK1) and thymidylate kinase (TMPK) (Hu et al., *J Biomed Sci.* 14, 491-7, 2007); Ke et al., *Genes Dev.* 19, 1920-33, 2005). This may partially explain why inhibition of RNR activity using various approaches reduces dTTP less than the other three types of dNTPs.

Sirt2 acts on RRM2 upstream deacetylase Sirt2 and indirectly affects RRM2 acetylation levels. In contrast, in **Fig.7b**, we re-express different acetylation mimetics including K95Q in RRM2 depleted cells in **Fig. 7**, which have direct constitutive acetylation of RRM2. K95Q mutation destroyed almost all RNR activity as shown in **Fig.1i** while silencing of Sirt2 only partially inhibits RNR activity as shown in **Fig.5d**. This may help explain why the K95Q mutation in **Fig. 7b** results in a different dNTP pool decrease pattern compared to new **Fig.5f**. We detected a greater reduction in dNTP levels in **Fig. 7b** than **Fig. 5f**.

If RNR activity is blocked by inhibition of SIRT2, we should expect to see an accumulation of cells in G1 phase. As shown in **new Fig. 5**, inhibition of Sirt2 by Sirt2 siRNA or Sirt2 inhibitor AGK2 caused RRM2 acetylation leading to decrease in RNR activity, reduction of dNTPs pool size and subsequent accumulation of cells in G1 phase. These results strongly suggest that reduction of RNR activity by acetylation via Sirt2 inhibition (siRNA or inhibitor) results in reduction of dNTPs pool size leading to accumulation of cells in G1 phase.

5. I am not sure if the title of the paper is very accurate ("Acetylation dictates ribonucleotide reductase function"). The function of RNR is to make dNTPs, and acetylation does not change it. The model proposes that acetylation inhibits ribonucleotide reductase activity.

Response: As suggested, we changed the title to: "Acetylation regulates ribonucleotide reductase activity and cancer cell growth".

We believe the substantial changes made have greatly strengthened the revised manuscript. We wish to thank the reviewers for their thoughtful and insightful comments. We hope to hear from you soon regarding the suitability of this revised manuscript for publication in "*Nature Communications*".

Reviewers' comments:

Reviewer #1 (Remarks to the Author):

The authors have very effectively addressed the previous critiques. Overall this is a well written paper with high quality data that shed new light on two proteins with important roles in cancer, RRM2 and SIRT2. I have a few minor suggestions related to wording as follows:

1. The authors have added mention of p53R2 in the Introduction in response to a previous comment. This is a positive addition but I would suggest the following minor changes to the text for clarity (p3, lines 14-17):

Additionally, p53R2 is encoded by the RRM2B gene, is induced by p53, and has been identified as a second radical-providing small subunit in mammalian cells¹⁴. The major role of p53R2-containing RNR complexes is in regulating the synthesis, replication and repair of mitochondrial DNA (mtDNA) in non-proliferating cells^{15, 16}

2. I hadn't noticed in Figure 4C the extent to which the expression level of the different sirtuins varies, which could impact the ability to detect interactions with RRM2. I don't dispute the conclusions drawn from the experiment but do think it would be appropriate to add a qualifier, along the lines of 'Although different sirtuins were overexpressed to different levels, only Sirt2 among seven sirtuin family members...' (p10, line 11).

3. The authors have added discussion of the Buisson et al. Mol. Cell paper that links ATR to RRM2 regulation. One key difference between this previous publication (as well as others such as Ref 17 cited by the authors) and the new work under consideration here is that the previous studies suggested indirect regulation of RRM2 by ATR via other kinases. Chen et al. suggest that ATR directly phosphorylates RRM2. However, these proteins are in different compartments (RRM2 in cytosol, ATR in nucleus), raising the question of how the proteins interact in the context of an intact cell. I'm not suggesting that any additional experiments need to be done, but this issue would be worth a mention in the discussion, perhaps with a comment about where the authors think the phosphorylation events are occurring (cytosol or nucleus).

4. A very minor point relates to figure 7F, which shows images of tumors from a xenotransplant experiment. For instances where no tumor is apparent, the authors have included dotted circles. These should be explained in the figure legend. Also, it isn't clear whether the different sizes and shapes of the dotted circles are meant to represent actual differences between samples.

Reviewer #2 (Remarks to the Author):

Revision of the manuscript "acetylation regulates ribonucleotide reductase activity and cancer cell growth" by Dr. Xingming Deng and colleagues.

Chen and co-workers present a manuscript that from my point of view strongly improved in this review process. They not only can resolve all of my concerns they also concisely answered open questions by performing several additional experiments. The results obtained from these experiments are all sound, the conclusions drawn are valid and strongly support the model presented. To highlight a few of these results, the authors could now concisely show, that the real lysine-acetylated RRM2 protein behaves as a monomer as does the RRM2 K to Q mutant. This was possible by producing site-specifically lysine-acetylated RRM2 protein using a synthetically evolved acyl-lysyl-tRNA-

synthetase/tRNACUA-pair from *Methanosarcina barkeri*. This on the one hand shows the mechanism underlying the observed impact of lysine-acetylation on RRM2 function. On the other hand-and this is even more important- it shows that in this case the RRM2 K to Q mutant protein can be used to mimic an lysine-acetylation event at the molecular level. This again supports the validity of all the in vivo experiments shown and the conclusions drawn, in which K to Q and K to R mutants of RRM2 proteins are used as mimics to study the impact on lysine-acetylation. Chen and colleagues furthermore studied the impact of individual inhibitors for classical and sirtuin deacetylases on RRM2 acetylation level, they show immunoblottings for successful knockdown of all deacetylases and acetyltransferases, they performed colocalisation studies of Sirt2 and RRM2 in G1 and S phase. Finally, they present a valid model on how acetylation could regulate RRM2 function in the physiological context. Overall, this study is an excellent and comprehensive report on how RNR is regulated by lysine-acetylation in the physiological context. Besides Chen and co-workers also mechanistically explain how RNR activity is regulated by lysine-acetylation. This study is a great contribution to the field and I strongly support the publication of this manuscript in its current version.

Reviewer #3 (Remarks to the Author):

In the revised manuscript, Xingming Deng and co-authors provide new data in an attempt to connect acetylation of the ribonucleotide reductase RRM2 subunit to regulation of dNTP pools. However, I find that the new data make the conclusions weaker and leave more questions than answers.

In Fig 1a and 1b, treatment of cells with NAM+TSA resulted in an ~80% drop of RNR activity and, as the authors state, "resulted in significantly decreased levels of all four dNTPs (Figure S1a and b)". Now at the request of reviewer 1 the authors have also have measured the cell cycle progression and found that "treatment of cells with TSA and NAM did not significantly alter the cell cycle (Supplementary Fig 1e)". This does not make sense as such huge decreases of RNR activity and dNTP pools should block the cell cycle, as the authors state themselves on p 13, line 4: "An adequate dNTP pool size is critical to ensure normal cell cycle progression and cell growth". In contrast, a ~40% drop of RNR activity observed in Fig 5e and a smaller decrease of dNTP pools (Fig 5f) compared to what was observed in Figure S1a and b as expected resulted in cell cycle arrest (Fig. 5g). These discrepancies indicate that dNTP pool measurements and/or RNR activity measurements, experiments technically fairly complicated, might not be correct. Indeed, a closer inspection of the raw dNTP data requested by reviewer 1 (see additional comment 3) makes me wonder why the retention times on of the dNTP standards (Supplementary Fig 1b) are completely different from the retention times of the dNTP samples (Supplementary Fig 1c). Of course, the retention times change as columns age, but this change is just too big to be sure about the identity of the peaks in the sample. It is also difficult to understand from the methods section whether dNTPs were normalized to anything and what HPLC column was used.

The authors claim several times that the investigated residues are highly conserved, underscoring their physiological importance, but this is misleading as the whole RRM2 protein is highly conserved. Homo sapiens and *Macaca nemestrina* RRM2 proteins are basically identical: out of 389 a.a. residues, only 5 are different. Even between the human and mouse RRM2 proteins there is 91% identity. If anything, the Fig 6g segment is less conserved than the surrounding sequence.

Score Expect Method Identities Positives Gaps Frame
744 bits(1922) 0.0() Compositional matrix adjust. 356/390(91%) 372/390(95%) 1/390(0%)

Features:

Query 1 MLSLRVPLAPITDPOQLQLSPLKGLSLVDKENTPPALSGTRVLASKTARRIFQEPTEPKT 60

MLS+R PLA I D QQLQLSPLK L+L DKENTPP LS TRVLASK ARRIFQ+ E ++
Sbjct 1 MLSVRTPLATIADQQQLQLSPLKRLTLADKENTPPTLSSTRVLASKAARRIFQDSAELES 60

Fig 1f segment

Query 61 KAAA-PGVEDEPLLRENPRRFVIFPIEYHDIWQMYKKAESFWTAEEDLSKDIOHWESL
119

KA P VEDEPLLRENPRRFV+FPYIEYHDIWQMYKKAESFWTAEEDLSKDIOHWE+L

Sbjct 61 KAPTNPVEDEPLLRENPRRFVVFPIEYHDIWQMYKKAESFWTAEEDLSKDIOHWEAL
120

Fig 6g segment

Query 120 KPEERYFISHVLAFFAASDGIVNENLVERFSQEVQITEARCFYGFQIAMENIHSEMYSL
179

KP+ER+FISHVLAFFAASDGIVNENLVERFSQEVQ+ TEARCFYGFQIAMENIHSEMYSL

Sbjct 121 KPDERHFISHVLAFFAASDGIVNENLVERFSQEVQVTEARCFYGFQIAMENIHSEMYSL
180

Query 180 IDTYIKDPKEREFLFNAIETMPCVKKKADWALRWIGDKEATYGERVVAFAAVEGIFSGS 239

IDTYIKDPKERE+LFNAIETMPCVKKKADWALRWIGDKEATYGERVVAFAAVEGIFSGS

Sbjct 181 IDTYIKDPKEREYLFNAIETMPCVKKKADWALRWIGDKEATYGERVVAFAAVEGIFSGS 240

Query 240 FASIFWLKKRGLMPGLTFSNELISRDEGLHCDFACLMFKHLVHKPSEERVREIINAVRI 299

FASIFWLKKRGLMPGLTFSNELISRDEGLHCDFACLMFKHLVHKP+E+RVREI NAVRI

Sbjct 241 FASIFWLKKRGLMPGLTFSNELISRDEGLHCDFACLMFKHLVHKPAEQRVREIITNAVRI 300

Query 300 EQEFLTEALPVKLIGMNCTLMKQYIEFVADRLMELGFSKVFVRVENPFDPMENISLEGKT 359

EQEFLTEALPVKLIGMNCTLMKQYIEFVADRLMELGF+K+FRVENPFDPMENISLEGKT

Sbjct 301 EQEFLTEALPVKLIGMNCTLMKQYIEFVADRLMELGFNKIFRVENPFDPMENISLEGKT 360

Query 360 NFFEKRVGEYQRMGVMSSPTENSFTLDADF 389

I accept that mutating an amino acid in a highly conserved protein could inhibit its dimer formation and lead to downregulation of RNR activity, decreased dNTP pools etc, but to what extent acetylation of this lysine in the native RRM2 regulates RNR activity in vivo is still not answered. In my point 3, I asked the authors to investigate what fraction of RRM2 is acetylated before and after TSA/NAM treatment. Is it for example 3% before TSA/NAM treatment, i.e. during normal cell cycle and 6% after, or is it for example 20% during normal cell cycle and 80% after TSA/NAM treatment? The latter would make their case stronger. Instead of answering this question, the authors replied that the acetylation sites were the same before and after TSA/NAM treatment.

Despite the overwhelming amount of data, in light of the discussed discrepancies, I do not find that the authors convincingly demonstrated that acetylation of RNR has physiological significance and do not recommend this manuscript for publication.

Response to Reviewer #1:

Reviewer #1 (Remarks to the Author):

The authors have very effectively addressed the previous critiques. Overall this is a well written paper with high quality data that shed new light on two proteins with important roles in cancer, RRM2 and SIRT2. I have a few minor suggestions related to wording as follows:

1. The authors have added mention of p53R2 in the Introduction in response to a previous comment. This is a positive addition but I would suggest the following minor changes to the text for clarity (p3, lines 14-17):

Additionally, p53R2 is encoded by the RRM2B gene, is induced by p53, and has been identified as a second radical-providing small subunit in mammalian cells¹⁴. The major role of p53R2-containing RNR complexes is in regulating the synthesis, replication and repair of mitochondrial DNA (mtDNA) in non-proliferating cells^{15, 16}

Response: We have revised these two sentences as recommended by Reviewer #1 in the revised text (page 3, lines 14-17).

2. I hadn't noticed in Figure 4C the extent to which the expression level of the different sirtuins varies, which could impact the ability to detect interactions with RRM2. I don't dispute the conclusions drawn from the experiment but do think it would be appropriate to add a qualifier, along the lines of 'Although different sirtuins were overexpressed to different levels, only Sirt2 among seven sirtuin family members...' (p10, line 11).

Response: As suggested, we have changed this sentence to "Although different sirtuins were overexpressed to different levels, only Sirt2 among seven sirtuin family members was observed to interact with RRM2 (Fig. 4c) in the revised text (page 11, lines 11-13).

3. The authors have added discussion of the Buisson et al. Mol. Cell paper that links ATR to RRM2 regulation. One key difference between this previous publication (as well as others such as Ref 71 cited by the authors) and the new work under consideration here is that the previous studies suggested indirect regulation of RRM2 by ATR via other kinases. Chen et al. suggest that ATR directly phosphorylates RRM2. However, these proteins are in different compartments (RRM2 in cytosol, ATR in nucleus), raising the question of how the proteins interact in the context of an intact cell. I'm not suggesting that any additional experiments need to be done, but this issue would be worth a mention in the discussion, perhaps with a comment about where the authors think the phosphorylation events are occurring (cytosol or nucleus).

Response: ATR is mainly located in the nucleus where it executes its DNA damage response

function (Kumar et al, Cell 2014, 158, 633-646). However, ATR has also been reported to be located in the cytoplasm (Cell Rep. 2017, 19, 1022-1032; Mol Cell 2015, 60, 35-46). In contrast, RRM2 is largely cytoplasmic (Nordlund et al, Annu. Rev. Biochem. 2006, 75, 681–706), but has also been reported to enter the nucleus during cell cycle progression or following CPT treatment (D'Angiolella et al, Cell. 2012, 149: 1023–1034; Zhang et al, J Biol Chem. 2009, 284:18085-95). Based on these reports, ATR and RRM2 could co-localize in the cytoplasm and nucleus. Our findings reveal that ATR not only interacts with RRM2 but also directly phosphorylates RRM2 at S150. Thus, ATR-mediated RRM2 phosphorylation may occur in both the cytoplasm and nucleus. This new discussion has been included in the revised text (**page 20, lines 6-12**).

4. A very minor point relates to figure 7F, which shows images of tumors from a xenotransplant experiment. For instances where no tumor is apparent, the authors have included dotted circles. These should be explained in the figure legend. Also, it isn't clear whether the different sizes and shapes of the dotted circles are meant to represent actual differences between samples.

Response: We have re-made figure 7f to use the same size and shape of dotted circles. All dotted circles indicate no visible tumors, which has been defined in the revised legend for figure 7f (**page 37, lines 18-19**).

Response to Reviewer # 2:

Reviewer #2 (Remarks to the Author):

Revision of the manuscript “acetylation regulates ribonucleotide reductase activity and cancer cell growth“ by Dr. Xingming Deng and colleagues.

Chen and co-workers present a manuscript that from my point of view strongly improved in this review process. They not only can resolve all of my concerns they also concisely answered open questions by performing several additional experiments. The results obtained from these experiments are all sound, the conclusions drawn are valid and strongly support the model presented. To highlight a few of these results, the authors could now concisely show, that the real lysine-acetylated RRM2 protein behaves as a monomer as does the RRM2 K to Q mutant. This was possible by producing site-specifically lysine-acetylated RRM2 protein using a synthetically evolved acyl-lysyl-tRNA-synthetase/tRNACUA-pair from Methanosarcina barkeri. This on the one hand shows the mechanism underlying the observed impact of lysine-acetylation on RRM2 function. On the other hand-and this is even more important- it shows that in this case the RRM2 K to Q mutant protein can be used to mimic an lysine-acetylation event at the molecular level. This again supports the validity of all the in vivo experiments shown and the conclusions drawn, in which K to Q and K to R mutants of RRM2 proteins are used as mimics to study the impact on lysine-acetylation. Chen and colleagues furthermore

studied the impact of individual inhibitors for classical and sirtuin deacetylases on RRM2 acetylation level, they show immunoblottings for successful knockdown of all deacetylases and acetyltransferases, they performed colocalisation studies of Sirt2 and RRM2 in G1 and S phase. Finally, they present a valid model on how acetylation could regulate RRM2 function in the physiological context. Overall, this study is an excellent and comprehensive report on how RNR is regulated by lysine-acetylation in the physiological context. Besides Chen and co-workers also mechanistically explain how RNR activity is regulated by lysine-acetylation. This study is a great contribution to the field and I strongly support the publication of this manuscript in its current version.

Response: We greatly appreciate Reviewer #2's strong support.

Response to Reviewer #3:

Reviewer #3 (Remarks to the Author):

In the revised manuscript, Xingming Deng and co-authors provide new data in an attempt to connect acetylation of the ribonucleotide reductase RRM2 subunit to regulation of dNTP pools. However, I find that the new data make the conclusions weaker and leave more questions than answers.

In Fig 1a and 1b, treatment of cells with NAM+TSA resulted in an ~80% drop of RNR activity and, as the authors state, "resulted in significantly decreased levels of all four dNTPs (Figure S1a and b)". Now at the request of reviewer 1 the authors have also have measured the cell cycle progression and found that "treatment of cells with TSA and NAM did not significantly alter the cell cycle (Supplementary Fig 1e)". This does not make sense as such huge decreases of RNR activity and dNTP pools should block the cell cycle, as the authors state themselves on p 13, line 4: "An adequate dNTP pool size is critical to ensure normal cell cycle progression and cell growth". In contrast, a ~40% drop of RNR activity observed in Fig 5e and a smaller decrease of dNTP pools (Fig 5f) compared to what was observed in Figure S1a and b as expected resulted in cell cycle arrest (Fig. 5g). These discrepancies indicate that dNTP pool measurements and/or RNR activity measurements, experiments technically fairly complicated, might not be correct.

Response: The treatment time of cells with NAM+TSA in old Supplementary Fig. 1e (18h) was different to the treatment time of cells with Sirt2 siRNA or AGK2 in Fig. 5g (48h). To avoid mistake, we repeated the experiment in supplementary Fig 1e by treating H1299 cells with TSA+NAM for a time course up to 48h, followed by analysis of cell cycle at various time points. Time course experiments indicate that a reduction in the S phase population was observed starting from the 18h time point after NAM+TSA treatment (i.e. 26.3% at 0h vs. 16.8% at 18h). The S phase population was continuously reduced to 4.6% at the 48h time point. These results indicate that NAM+TSA treatment can reduce the S phase population in a time-dependent manner. These results support the data in Fig. 5g showing that treatment of cells with Sirt2

siRNA or Sirt2 inhibitor AGK2 for 48h resulted in reduction of the S phase population. Thus, TSA/NAM-induced inhibition of RNR activity may reduce the cell population in S phase. Additionally, we also discovered that NAM+TSA treatment induced a time-dependent G2/M arrest starting from 12h. The exact mechanism(s) is not clear. Several mitotic proteins (i.e. BubR1, NudC, Aurora B and histones) have been reported to be regulated by acetylation (EMBO J. 2009, 28: 2077-89.; PLoS One. 2013, 8: e73841; FASEB J. 2012, 26: 4057-67. Mol Biol Cell. 2003, 14: 3821-33), which may be related to NAM+TSA-induced G2/M arrest via acetylation. However, this is beyond the scope of this manuscript. These new data have been incorporated into Supplementary Fig. 1e and f, and described in the revised text (**page 5, lines 21-24; page 6, lines 1-3**).

Indeed, a closer inspection of the raw dNTP data requested by reviewer 1 (see additional comment 3) makes me wonder why the retention times on of the dNTP standards (Supplementary Fig 1b) are completely different from the retention times of the dNTP samples (Supplementary Fig 1c). Of course, the retention times change as columns age, but this change is just too big to be sure about the identity of the peaks in the sample. It is also difficult to understand from the methods section whether dNTPs were normalized to anything and what HPLC column was used.

Response: Regarding the discrepancies in the retention times between old Fig. S1b and Fig. S1c, the experiments shown in old Supplementary Fig 1b and Fig. 1c used different columns at different times in the core facility. This was an error. For old Supplementary Fig. 1b, the standard from 8/16/2016 was run on a 250mm column. For Supplementary Fig. 1c, samples from 10/11/2016 were run on a 150mm column. Now, we have corrected this error, and add the standard profile from the same experiments using the same type of column during experiments performed on 10/11/2016 for Fig. S1c as shown in revised Supplementary Fig. 1b and c on the right of this paragraph. The HPLC column we used on 10/11/2016 was C18 3.5 μ m (150 \times 4.6mm) column (Waters Corporation, Milford, MA), and this information has been included in “Methods” in the revised text (**page 24, lines 19-21**). This new data for “Standard” has been incorporated into Supplementary Fig. 1b.

The authors claim several times that the investigated residues are highly conserved, underscoring their physiological importance, but this is misleading as the whole RRM2 protein is highly conserved. Homo sapiens and Macaca nemestrina RRM2 proteins are basically identical: out of 389 a.a. residues, only 5 are different. Even between the human and mouse RRM2 proteins there is 91% identity. If anything, the Fig 6g segment is less conserved than the surrounding sequence.

Score Expect Method Identities Positives Gaps Frame
744 bits(1922) 0.0() Compositional matrix adjust. 356/390(91%) 372/390(95%) 1/390(0%)

Features:

Query 1

MLSLRVPLAPITDPQQQLQLSPLKGLSLVDKENTPPALSGTRVLASKTARRIFQEPTPKT 60
MLS+R PLA I D QQLQLSPLK L+L DKENTPP LS TRVLASK ARRIFQ+ E ++

Sbjct 1

MLSVRTPLATIADQQQLQLSPLKRLTLADKENTPPTLSSTRVLASKAARRIFQDSAELES 60

Fig 1f segment

Query 61 KAAA-

PGVEDEPLLRENPRRFVIFPIEYHDIWQMYKKA**EAS**FWTAEVLDLSKDIQHWESL 119
KA P VEDEPLLRENPRRFV+**FPIEYHDIWQMYKKA**EAS****FWTAEVLDLSKDIQHWE+L

Sbjct 61

KAPTNPVVEDEPLLRENPRRFVVFPIEYHDIWQMYKKA**EAS**FWTAEVLDLSKDIQHWEAL
120

Fig 6g segment

Query 120

KPEERYFISHVLAFFAASDGIVNENL**VERFSQEVQ**ITEARCFYGFQIAMENIHSEMYSL 179
KP+ER+FISHVLAFFAASDGIVNENL**VERFSQEVQ**+TEARCFYGFQIAMENIHSEMYSL

Sbjct 121

KPDERHFISHVLAFFAASDGIVNENL**VERFSQEVQ**VTEARCFYGFQIAMENIHSEMYSL 180

Query 180

IDTYIKDPKEREFLFNAIETMPCVKKKADWALRWIGDKEATYGERVVAFAAVEGIFFSGS 239
IDTYIKDPKERE+LFNAIETMPCVKKKADWALRWIGDKEATYGERVVAFAAVEGIFFSGS

Sbjct 181

IDTYIKDPKEREYLFNAIETMPCVKKKADWALRWIGDKEATYGERVVAFAAVEGIFFSGS 240

Query 240

FASIFWLKKRGLMPGLTFSNELISRDEGLHCDFACL MFKHLVHKPSEERVREII NAVRI
299 FASIFWLKKRGLMPGLTFSNELISRDEGLHCDFACL MFKHLVHKP+E+RVREII

NAVRI

Sbjct 241

FASIFWLKKRGLMPGLTFSNELISRDEGLHCDFACL MFKHLVHKPAEQRVREIITNAVRI 300

Query 300

EQEFLTEALPVKLIGMNCTLMKQYIEFVADRLMLELGF SKVFRVENPFDFMENISLEGKT 359
EQEFLTEALPVKLIGMNCTLMKQYIEFVADRLMLELGF+K+FRVENPFDFMENISLEGKT

Sbjct 301

EQEFLTEALPVKLIGMNCTLMKQYIEFVADRLMLELGFNKIFRVENPFDFMENISLEGKT 360

Query 360 NFFEKRVGEYQRMGVMSSPTENSFTLDADF 389

Response: We agree with reviewer #3's point regarding the wording "highly conserved". We have revised our description in the text by removing the sentence "... is highly conserved across various species" from page 7 and page 15. We have deleted the old Fig. 1f and revised Fig. 6g accordingly.

I accept that mutating an amino acid in a highly conserved protein could inhibit its dimer formation and lead to downregulation of RNR activity, decreased dNTP pools etc, but to what extent acetylation of this lysine in the native RRM2 regulates RNR activity in vivo is still not answered. In my point 3, I asked the authors to investigate what fraction of RRM2 is acetylated before and after TSA/NAM treatment. Is it for example 3% before TSA/NAM treatment, i.e. during normal cell cycle and 6% after, or is it for example 20% during normal cell cycle and 80% after TSA/NAM treatment? The latter would make their case stronger. Instead of answering this question, the authors replied that the acetylation sites were the same before and after TSA/NAM treatment.

Response: To measure the percentage of acetylated RRM2 (Ac-K RRM2) before and after NAM/TSA treatment in cells, we used Ac-K immunoaffinity beads (Cell Signaling Technologies, Cat# 13416) to deplete Ac-K proteins, including Ac-K RRM2, from cell lysates isolated from H460 and BEAS-2B cells before and after TSA/NAM treatment as previously described (J Biol Chem. 2002 Dec 27;277(52):50607-11), followed by Western blot analysis of unacetylated RRM2 using anti-RRM2 antibody and quantifying the unacetylated RRM2 on Western blot bands using ImageJ software. The percentage of Ac-K RRM2 was calculated using the formula: % Ac-K RRM2 = (Total RRM2 - Unacetylated RRM2)/Total RRM2 ×100 as indicated in Supplementary Fig. 2a. To test whether Ac-K RRM2 can be depleted from lysates by Ac-K immunoaffinity beads, we measured Ac-K RRM2 by IP using Ac-K specific antibody in the lysates before vs. after Ac-K depletion, followed by Western blot analysis of Ac-K RRM2 using anti-RRM2 antibody. Before Ac-K depletion, certain levels of Ac-K RRM2 were observed in H460 and BEAS-2B cells, and NAM/TSA enhanced Ac-K RRM2. However, no detectable levels of Ac-K RRM2 were observed in the lysates after Ac-K depletion (Supplementary Fig. 2b), indicating a highly efficient depletion of Ac-K RRM2 from the lysates by Ac-K immunoaffinity beads. To obtain the percentages of Ac-K RRM2, we measured total RRM2 and unacetylated RRM2 in the lysates before and after Ac-K depletion in H460 and BEAS-2B cells with and without NAM/TSA treatment. After quantification and calculation using the formula described above, we found that 30% and 26% of RRM2 was acetylated before NAM/TSA treatment in H460 and BEAS-2B cells, respectively (Supplementary Fig. 2c). After NAM/TSA treatment, 76% and 68% of RRM2 was acetylated in H460 and BEAS-2B cells, respectively (Supplementary Fig. 2c). These results indicate that NAM/TSA can enhance RRM2 acetylation. These new data have been included as new **Supplementary Fig. 2**, and described in the revised text (**page 6, lines 9-24; page 7, lines 1-6**).

Despite the overwhelming amount of data, in light of the discussed discrepancies, I do not find that the authors convincingly demonstrated that acetylation of RNR has physiological significance and do not recommend this manuscript for publication.

Response: Since we have addressed all concerns from all reviewers, including those of Reviewer#3, we hope this second revision will satisfy the reviewers, and thank all Reviewers for their additional and valuable comments.

We believe the substantial changes made in the second revision have greatly strengthened this manuscript. We wish to thank all reviewers for their thoughtful and insightful comments. We hope to hear from you soon regarding the suitability of this second revised manuscript for publication in "*Nature Communications*".

Reviewers' comments:

Reviewer #3 (Remarks to the Author):

In the second revision, Xingming Deng and co-authors provide additional data in an attempt to connect acetylation of the ribonucleotide reductase RRM2 subunit to regulation of dNTP pools. Unfortunately, I am now even less convinced that acetylation of RRM2 plays any significant role in regulation of its activity and by extension the cell cycle regulation.

The authors provide new data in Supplementary Fig. 1e where in time-course experiments they observe "a reduction in the S phase population starting from the 18h time point after NAM+TSA treatment (i.e. 26.3% at 0h vs. 16.8% at 18h). The S phase population was continuously reduced to 4.6% at the 48h time point. These results indicate that NAM+TSA treatment can reduce the S phase population in a time-dependent manner." They authors write that "these results support the data in Fig. 5g showing that treatment of cells with Sirt2 siRNA or Sirt2 inhibitor AGK2 for 48h resulted in reduction of the S phase population. Thus, TSA/NAM-induced inhibition of RNR activity may reduce the cell population in S phase."

This time-course experiment and the conclusion do not make sense considering everything we know about RNR inhibition. The effect should be exactly the opposite: when RNR is inhibited and dNTP pools are reduced, the cells should first start to accumulate in S phase, because the replication slows down. See for example Fig 1 in the landmark paper from Brewer and Raghuraman, which shows how an RNR inhibitor hydroxyurea causes S phase to proceed in slow motion:

<https://mcb.asm.org/content/27/18/6396>

After a prolonged inhibition of dNTP synthesis the cells might eventually finish the S phase and accumulate at the next G1 or G1/S boundary, and I thought that this was the effect observed in Fig 5g as this figure shows a 48 hours NAM-TSA treatment, but now with the new and more detailed time-course experiments I realize that the authors never observe an expected accumulation of cells in S phase, which strongly suggests that there is no RNR inhibition in their experiments. Therefore, in my opinion the whole model falls apart.

It is also questionable that an increase in RRM2 acetylation from the basal 26-30% level to 68-76% level after the NAM/TSA treatment could lead to an 80% drop in RNR activity.